


# Deep learning for the estimation of water-levels using river cameras

Remy Vandaele[1,3], Sarah L. Dance[1,2], and Varun Ojha[3]

[1]Department of Meteorology, University of Reading, U.K
[2]Department of Mathematics, University of Reading, U.K
[3]Department of Computer Sciences, University of Reading, U.K

**Correspondence:** Remy Vandaele (r.a.vandaele@reading.ac.uk)

**Abstract.** River level estimation is a critical task required for the understanding of flood events, and is often complicated by the scarcity of available data. Recent studies have proposed to take advantage of large networks of river camera images to estimate the river levels, but currently, the utility of this approach remains limited as it requires a large amount of manual intervention (ground topographic surveys and water image annotation). We develop an approach using an automated water

semantic segmentation method to ease the process of river level estimation from river camera images. Our method is based on the application of a transfer learning methodology to deep semantic neural networks designed for water segmentation. Using datasets of image series extracted from four river cameras and manually annotated for the observation of a flood event on the Severn and Avon rivers, UK (21 November - 5 December 2012), we show that our algorithm is able to automate the annotation process with an accuracy greater than $91\%$. Then, we apply our approach to year-long image series from the same

cameras observing the Severn and Avon (from 1 June 2019 to 31 May 2020) and compare our results with nearby river-gauge measurements. Given the high correlation (Pearson's Correlation Coefficient $> 0.94$) between our results and the river-gauge measurements, it is clear that our approach to automation of the water segmentation on river camera images could allow for straightforward, inexpensive observation of flood events, especially at ungauged locations.

## 1 Introduction

Fluvial flood forecasting systems often deploy hydrodynamic inundation models to compute water-level and velocity in the river, and, when the storage capacity of the river is exceeded, in the floodplain (e.g., Flack et al., 2019). Simulation library approaches using pre-computed hydrodynamic model solutions are also becoming more common for near-real-time flood mapping (e.g., Speight et al., 2018). Observations of fluvial floods are key to model improvement, both to improve forecasts during the event via data assimilation (e.g., Ricci et al., 2011; García-Pintado et al., 2013, 2015; Di Mauro et al., 2020;

Cooper et al., 2019), and to identify model shortcomings and improvements in post-event analysis (e.g., Werner et al., 2005). Water-level observations are often easier to obtain than streamflow observations, as they do not require any information about the rating curve. Furthermore, several studies have demonstrated their utility for calibration of hydrological models (e.g. van Meerveld et al., 2017; Seibert and Vis, 2016).

The main types of water-level observations possible with current technologies include ground-based and remote-sensing

techniques. River-gauges allow continuous monitoring of river levels at point locations. However, their measurements may



not be valid if the gauge is overwhelmed in an extreme flood. The network of river gauging stations is declining globally (Vörösmarty et al., 2001). In consequence, many flood-sensitive areas are ungauged, or must be studied through river-gauges that can be located several kilometers away (e.g., Neal et al., 2009), and so they cannot accurately describe the local situation.

Satellite and airborne images can be used to derive flood extents, and, when combined with a digital elevation model (DEM),
water-levels along the flood edge (Grimaldi et al., 2016). These images can be obtained using optical sensors, or synthetic aperture radar (SAR). Satellite and airborne optical techniques are hampered by their daylight-only application and their inability to map flooding beneath clouds and vegetation (Yan et al., 2015). On the other hand, SAR images are unaffected by cloud and can be obtained day or night. Thus, their use for flood mapping in rural areas is well established (e.g., Mason et al., 2012; Giustarini et al., 2016). In urban areas, shadow and layover issues make the flood mapping more challenging (e.g., Mason et al.,
2018; Tanguy et al., 2017). In addition, SAR satellite overpasses are infrequent (at most once or twice per day, depending on location), so it is uncommon to capture the rising limb of the flood (Grimaldi et al., 2016).

Unmanned aerial systems (UAS) are an emerging technology increasingly being used for river observations (Tauro et al., 2018). However, UAS deployment is subject to civil aviation restrictions (e.g., Civil Aviation Authority, 2020). Furthermore, there is a balance between instrument payload and the need to land and refuel. Images are subject to UAS drift and require
complex orthorectification (Perks et al., 2016).

There have been a number of citizen science projects that investigated the use of crowdsourced observations of river level (e.g. Lanfranchi et al., 2014; Etter et al., 2020; Lowry et al., 2019; Walker et al., 2019; Baruch, 2018). However, in this paper, we use "opportunistic data" (Hintz et al., 2019) from an existing network of river cameras to observe flood events. River cameras typically continuously broadcast live images from waterways. The cost of installation and maintenance of such cameras is low
as they only rely on the availability of electricity through a power grid or (back-up) batteries, and the upload of the images can be organised through standard and/or mobile broadband. Many of these cameras are installed at ungauged locations (Vetra-Carvalho et al., 2020b; Perks et al., 2020; Lo et al., 2015), and they have become a common tool for the monitoring of rivers for many private (fishing, tourism, boating,...) and public (flood prevention, river management) purposes. Thus, the use of existing cameras could offer a good coverage of the river network.

By extracting the location of water-filled pixels from a stream of river camera images (water segmentation), it becomes possible to analyse flood events happening within the field-of-view of a camera. Most attempts that have tried to tackle the problem of automated water detection in the context of floods have been realised using hand-crafted features (Filonenko et al., 2015). These algorithms remain sensitive to luminosity and water reflection problems (Filonenko et al., 2015). Deep learning approaches have been applied to flood detection in (Lopez-Fuentes et al., 2017; Moy de Vitry et al., 2019). However, current
flood-related studies using river camera images are limited because the observations made on the stream of images must be annotated manually (Vetra-Carvalho et al., 2020b). An accurate, manual annotation of such images is a long and tedious process that compels the analyst to narrow the scope (number of images considered) of the study.

In this paper, we study a deep learning algorithm allowing the automated segmentation of water pixels on river camera images. We show its relevance and usage in the context of flood-related studies. The algorithm that we develop in this study is
based on the use of deep semantic segmentation networks and transfer learning to compensate for the lack of annotated images.





The contributions we bring with this paper are the following:

– We develop a tool for the study of floods using river camera images. This method is based on the application of a transfer learning methodology to deep semantic segmentation networks in order to repurpose them for the segmentation of water.

– We study the performance on two-week datasets of manually annotated river camera images extracted from four river cameras monitoring the flood of the Severn or the Avon rivers in the Tewkesbury area, in the UK, between 21 November and 5 December 2012.

– On a much larger dataset that consists of a year-long image series from the same four camera locations (from 1 June 2019 to 31 May 2020), but without any available river-level annotations, we show that our method is able to track water-level evolution during floods and while the river is in-bank.

In Section 2, we describe the current techniques employed for automated water level estimation, and introduce concepts that allowed the development of our own automated water segmentation method. In Section 3, we detail the approach that we used to develop our water-level estimation approach. In Section 4, we present and analyse the results of our experiments. Finally, we conclude in Section 5.

## 2 Deep learning for water segmentation and water-level estimation

This background section is divided into three parts. In the first part, we introduce the problem of the automation of water-level estimation from camera images using water segmentation. In the second part, we briefly introduce the current techniques used for semantic segmentation based on the use of convolutional neural networks, and we discuss the main results from the literature specific to water semantic segmentation. In the third part, we explain the concept of transfer learning that we used in our method.

### 80 2.1 Water segmentation for water-level estimation

In this work, we tackle the problem of river level estimation through the use of automated **semantic segmentation** algorithms applied to river camera images. We focus on automated river/water semantic segmentation. As we show in Fig. 1, a water semantic segmentation algorithm will associate a Boolean variable 1 (flooded)/ 0 (unflooded) to each pixel of an RGB image, expressing whether or not there is water present in the pixel. The Boolean mask will thus have as many pixels as the RGB 85 image.

Producing an automated water segmentation algorithm is a major milestone in order to use river camera images for river level estimation. No matter whether additional metadata (such as ground-survey or DEM) is available or not, a generic water-level estimation algorithm will need to determine which pixels of the images are flooded, and which are not (Grimaldi et al., 2016).

For example, without any type of metadata regarding the height of locations within the field-of-view of the camera, a first 90 generic approach presented by Moy de Vitry et al. (2019) proposes to use the percentage of water pixels in the image to observe the relative evolution of the water-level.





Original image

Water segmentation mask

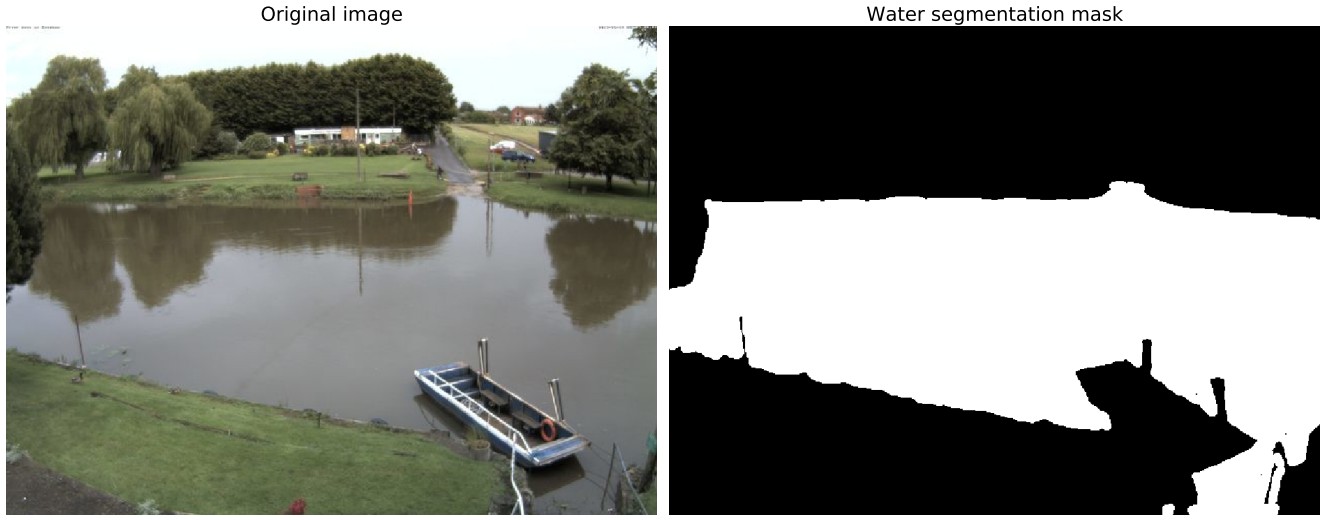

**Figure 1.** Example of a water segmentation mask (right) for a river camera image (left). The mask corresponds to a pixel-wise labelling of the original images between flooded pixels (in white) and unflooded pixels (in black), expressing whether or not there is water present in the pixel.

## 2.2 Deep learning for automated water segmentation

In this section, we review current perspectives on the automation of water segmentation to provide the background for our choice of method.

As for most image-processing related tasks, recent advances in optimisation, parallel computing and dataset availability have allowed deep learning methods, and specifically deep convolutional neural networks (CNNs) to bring major improvements to the field of automated semantic segmentation (Guo et al., 2018). CNNs are a type of neural networks where input images are processed through convolution layers. As we show in Fig. 2, with convolutional neural networks, an image is divided into square sub-regions (tiles) of size $F \times F$ that can possibly overlap. The image is processed through a series of convolutional

layers. A convolutional layer is composed of filters (matrices) of size $F \times F$. For each filter of the convolutional layer, the filter is applied on each of the tiles of the image by computing the sum of the Hadamard product (element-wise matrix multiplication) - also called a convolution in deep learning - between the tile and the filter (Strang, 2019). If the products of the convolution operations are organised spatially, the output of a convolutional layer can be seen as another image which itself can be processed by another convolutional layer. During the training of the networks, the weights of the filters (the matrix values) are optimised.



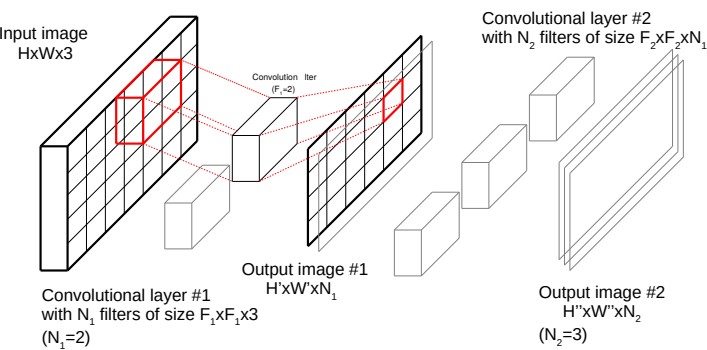

**Figure 2.** Example of convolution layers inside a neural network.

The idea is that the filters will converge along the convolutional layers towards weights making the input image more and more meaningful for the task at hand.

    Currently, the availability of large datasets with pixel-wise annotated images such as COCO-stuff (Caesar et al., 2018), ADE20k (Zhou et al., 2017), Cityscapes (Cordts et al., 2016), PASCAL VOC (Everingham et al., 2010) have popularised the development and comparison of semantic segmentation networks, a type of CNN designed for semantic segmentation. Those

networks have achieved impressive results (Caesar et al., 2018; Zhou et al., 2017; He et al., 2017). In our work, we will use the datasets COCO-stuff and ADE20k (see Section 3).

    An example of the use of deep learning applied to flood detection is the work of Lopez-Fuentes et al. (2017). The authors performed water segmentation on a home-made, accessible, dataset of 300 water images that were gathered from the web and annotated manually. The performance of three semantic segmentation networks (FCN-8 Long et al. (2015), Tiramisu Jégou

et al. (2017) and Pix2Pix Isola et al. (2017)) was evaluated. By training the networks from scratch (randomly initialised filter weights), Tiramisu produced the best results, with 90.47% pixel accuracy. We will use the dataset developed by Lopez-Fuentes et al. (2017) in our study (see Section 3).

    In Moy de Vitry et al. (2019), the authors also proposed to use a deep semantic segmentation network trained from scratch to produce a generic algorithm for flood level trend monitoring. The biggest originality of this paper lies in their development of

the SOFI index that corresponds to the percentage of pixels estimated as water pixels by the network in the image. This index allows the authors to monitor the water-levels evolution on their datasets. In our experiments, we will use the SOFI index to track water-level changes (see Section 4).

### 2.3   Transfer learning

Inductive transfer learning (TL) is commonly used to repurpose efficient machine learning models trained on large datasets of

well-known problems in order to address related problems with smaller training datasets. Indeed, typically, water segmentation





networks are trained on small datasets composed of 100-300 training images (Lopez-Fuentes et al., 2017; Steccanella et al., 2018; Moy de Vitry et al., 2019) while more popular problems can be trained on datasets composed of more than 15000 images (e.g, Caesar et al. (2018); Zhou et al. (2017)). In many cases, using inductive TL approaches for the training of CNNs instead of training them from scratch, with randomly initialised weights, allows improvement in the network performance (Reyes et al.,
2015; Sabatelli et al., 2018).

For a typical supervised machine learning problem, the aim is to find a function $f : X \rightarrow Y$ from a dataset $B = \{(x_i, y_i)_{i=1}^N : x_i \in X, y_i \in Y\}$ of $N$ input-output pairs such that the function $f$ should be able to predict the output of a new (possibly unseen) input, as accurately as possible. The set $X$ is called the input space, and $Y$ the output space.

With TL, the aim is to also build a function $f_t : X_t \rightarrow Y_t$ for a *target* problem with input space $X_t$, output space $Y_t$ and a
dataset $B_t$. TL tries to build $f_t$ by *transferring* knowledge from a *source* problem $s$ with input space $X_s$, output space $Y_s$ and a dataset $B_s$.

Inductive TL (Pan and Yang, 2009) is the branch of TL which is related to problems where datasets of input-output pairs are available in both source $(X_s, Y_s)$ and target $(X_t, Y_t)$ domains, and where the source and target input spaces are similar $(X_s \approx X_t)$ but not the output space $(Y_s \neq Y_t)$.
The specific approach that we used to apply TL is presented in Section 3.

## 3   Methodology

In this section, we present our approach as well as the different techniques and materials related to its development. Note that a part of our water semantic segmentation approach was presented in Vandaele et al. (2020a). Here, we provide a perspective centered around the application of our method in hydrology. We apply our method on new relevant datasets, and evaluate its
relevance in the context of water-level estimation. All the results presented in this paper are novel.

### 3.1   Application of transfer learning

In the context of water semantic segmentation using a CNN, we want to find the best possible filter values for the convolutional neural network ($f$) by using a set of training images ($X$) associated with their water mask ($Y$) so that the network will be able to predict the water mask of a new image as accurately as possible.
In this work, we use an inductive TL approach to apply TL to CNNs: we first train a CNN on the large dataset of a *source* problem in order to get its filter weight values. Then, we fine-tune the weights of the networks over a smaller *target* problem, which means that the filter values obtained during the training on the source problem are used as initial values for the training of the target problem.

However, if we want to apply this approach, we then need to make several choices:

1. The semantic segmentation network architecture(s) that we are going to use.

2. The source problem dataset(s) on which the networks will first be trained.





| ADE20k dataset | | COCO-stuff dataset | |
| --- | --- | --- | --- |
| Labels | # images | Labels | # images |
| water | 709 | river | 2113 |
| sea | 651 | sea | 6598 |
| river | 320 | water-other | 2453 |
| waterfall | 80 | | |

**Table 1.** Labels related to water bodies, and the number of images that contain at least one pixel with the corresponding label.

3. The target problem dataset(s) on which the networks will be fine-tuned.

4. The configuration of the fine-tuning procedure(s)

For the purpose of this study, we chose to consider several options regarding these choices. These are outlined in the rest of
the section.

## 3.2 Network architectures and source datasets

For this study, we considered two state-of-the-art CNNs for semantic segmentation (semantic segmentation networks):

The first network that we considered is **ResNet50-UperNet (RU)**. This network is an UperNet network with a ResNet50
image classification network used as a backbone. In our case, ResNet50-UperNet was trained on the ADE20k dataset (Zhou
et al., 2018). ResNet50 (He et al., 2016) is a typical CNN architecture used for image classification tasks (at the image level)
that the UperNet architecture transforms into a semantic segmentation network. ADE20k is a dataset designed for indoor and
outdoor scene parsing, with 22000 images semantically annotated with 150 labels, among which 4 are water related labels (see
Table 1).

**DeepLab (v3)** is the second network that was considered. This network was trained and has produced state-of-the-art results
on the COCO-stuff dataset (Chen et al., 2017). DeepLab also uses a ResNet50 network as a backbone network, but performs
the upsampling of the backbone's last layers by using atrous convolutions (Chen et al., 2017). COCO-stuff is a dataset made
of 164000 images semantically annotated with 171 labels, among which 3 are related to water objects (see Table 1).

## 3.3 Target datasets for water semantic segmentation

In order to fine-tune the networks trained on the source problems, we considered two different target datasets:

– **LAGO** (named after the first author of the study presented in Lopez-Fuentes et al. (2017)), is a dataset of RGB images
with binary semantic segmentation of water/not-water masks. The dataset was created through manual collection of
camera images having a field-of-view capturing riverbanks. The big advantage of this dataset is that the images are
directly for river segmentation (Lopez-Fuentes et al., 2017). It is a dataset made of 300 images, with 225 used in training.





- **WATERDB** is a dataset of RGB images with binary semantic segmentation of water/not-water labelled pixels that we
created through the aggregation of images containing label annotations related to water bodies coming from the ADE20k
(Zhou et al., 2017) (water, sea, river, waterfall) and the COCO-stuff (Caesar et al., 2018) (river, sea, water-other) dataset
(see Table 1). The dataset is made of 12684 training images.

While LAGO is a dataset that is more directly related to the segmentation of river-camera images, it is also a dataset with a
much smaller set of images than WATERDB. By choosing these two datasets, we can determine if we have better results when
we fine-tune the networks over large datasets with images that are not always directly related to the segmentation of water
on river-camera images, or conversely if we obtain better results by fine-tuning the networks over smaller but more relevant
datasets.

### 3.4 Fine-tuning the networks

In Section 3.2, we chose semantic segmentation networks that are addressing semantic segmentation problems with 171
(COCO-stuff) and 150 (ADE20k) labels. However, our water semantic segmentation problem is a binary segmentation prob-
lem, with only two labels: water or not-water. In practice, this means that the dimensions of the last output layer of the *source*
semantic segmentation networks and the *target* semantic segmentation networks might not be of the same size. In consequence,
it is not possible to use the weights of the last layer of the source network to initialise the weights of the last layer of the target
network. This is why we considered two fine-tuning strategies in this study:

- **WHOLE.** We fine-tune the entire target network with all the initial weights of all layers equal to the weights of the
source network except for a random initialisation of the last binary output layers.

- **2STEPS.** We first retrain the last layer of the target network (with random initialisation) with all the other layers frozen
to the weights of the source network layers. Once the last layer is retrained, we fine-tune the entire target network.

We chose to investigate the behaviour of different types of CNNs: not only in terms of architecture, but also in terms of
the dataset preparation and network fine-tuning strategies. In particular, we have analysed the behaviour of two network ar-
chitectures pre-trained over specific datasets (ResNet50-UperNet pre-trained over ADE20k and DeepLab (v3) pre-trained over
COCO-stuff) and two fine-tuning strategies (WHOLE or 2STEPS) applied on two different datasets (LAGO or WATERDB).
This means that 8 different network configurations were trained (see Fig. 3).

The details regarding the specific parameter settings used for the training of the source and target network can be found in
Vandaele et al. (2020a), where we showed that we were able to reach pixel accuracy $> 96\%$ during segmentation.

## 4 Experiments

We carried out two experiments with the approach that we presented in Section 3. The first experiment presented in Section
4.1 is designed to provide quantifiable results using datasets annotated with landmarks and water-level information, used in





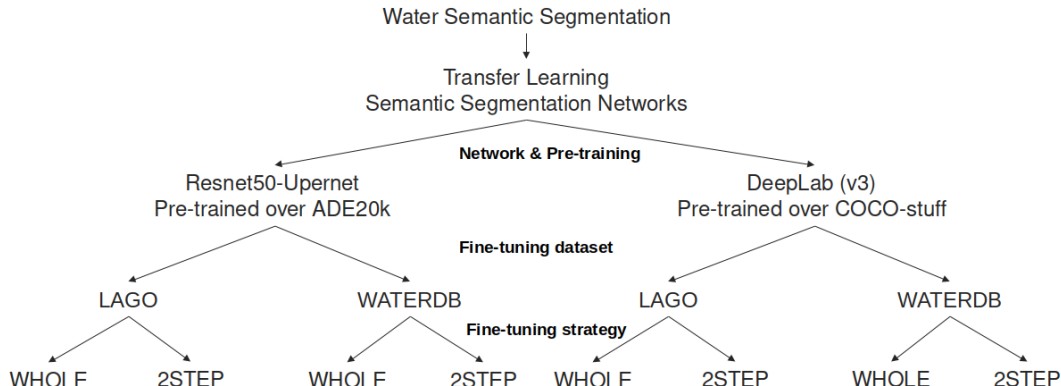

**Figure 3.** Model configurations used with our TL methodology

practice for the observation of a flood event (Vetra-Carvalho et al., 2020b). The results show the suitability of our approach for
flood extent analysis using river cameras, and provide an initial estimate of the water-levels using the landmarks. However, the
annotated datasets used for the first experiment were relatively small, as they consist of two-week image series focusing on a
specific flood event.

In the second experiment, presented in Section 4.2, we evaluated the performance of our water semantic segmentation
algorithm over images coming from a 12-month image time-series. With this dataset, we show that our method is also able
to track different flood events observed using the same river cameras, as well as to track smaller changes in water-level. As
this dataset was not annotated by a human observer, we compared our results with river level data obtained from the nearest
available river-gauges.

## 4.1   Application on a practical case for flood observation

### 4.1.1   River camera datasets for a flood event on the river Severn and the river Avon

For this experiment, we considered 4 different cameras located along the Severn and Avon rivers, UK at Diglis Lock (DIGL),
Tewkesbury Marina (TEWK), Strensham Lock (STRE), and Evesham (EVES). The images capture a major flood event that
occurred in the Tewkesbury area between 21 November and 5 December 2012. This is a well-observed and well-studied event
(García-Pintado et al., 2015). Further information about the camera locations can be found in Vetra-Carvalho et al. (2020b).

The cameras are part of the Farson Digital Watercams (https://www.farsondigitalwatercams.com/) network. The field-of-
view of the cameras stays fixed (no camera rotation or zoom). The images have been captured using a Mobotix M24 all-purpose
high-definition (HD) web-camera system with 3MP (megapixels) producing 2048 × 1536-pixel RGB images. The images at
our disposal were all watermarked, but a visual inspection of our results showed that those watermarks had near to no influence
on the segmentation performance.





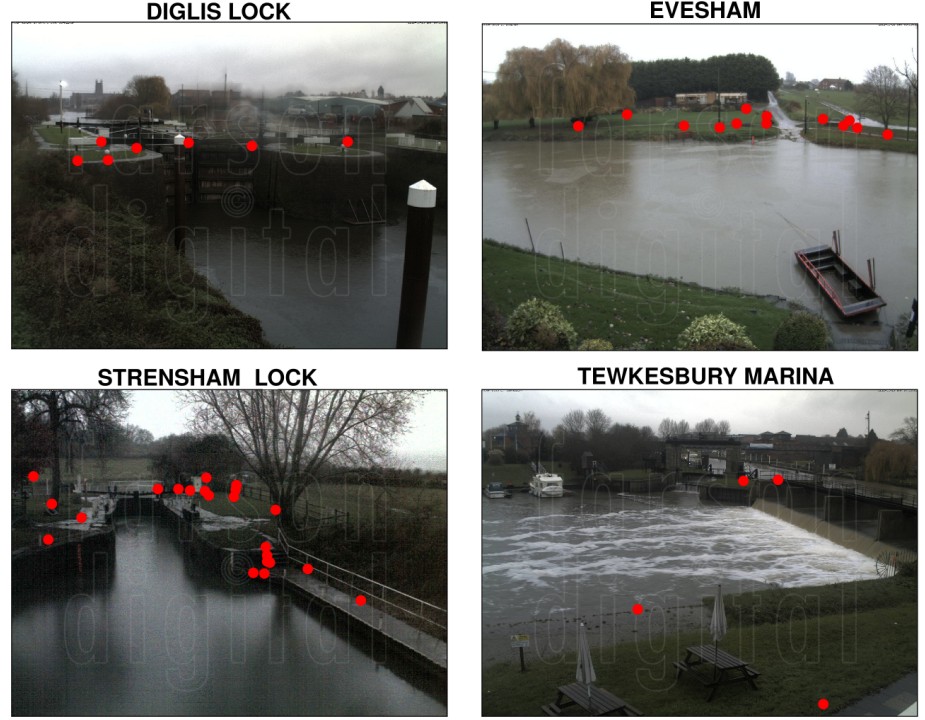

**Figure 4.** Sample camera image for each location, with the measured landmarks annotated by red dots.

| Dataset name | Location | #images | # landmarks | % flooded landmarks | Camera location (Northing, Easting) |
|---|---|---|---|---|---|
| DIGL | Diglis Lock | 141 | 7 | 24.11 | (253402.08m, 384691.15m) |
| EVES | Evesham | 134 | 13 | 30.94 | (243656.21m, 402923.2m) |
| STRE | Strensham Lock | 144 | 24 | 37.15 | (240449.13m, 391564.37m) |
| TEWK | Tewkesbury Marina | 138 | 4 | 43.66 | (233394.44m, 389466.95m) |

**Table 2.** River-camera location and specific dataset information.

For each camera, ground surveys have previously been in order to measure the topographic height of several landmarks

within the field-of-view of the camera (Vetra-Carvalho et al., 2020b). Note that the number and spread of measured landmarks over the camera's field-of-view was constrained to locations that were accessible during the ground survey. For each camera, daytime hourly images (around 9 per day) were retrieved and annotated by a human-observer using the surveyed landmarks as a reference in order to estimate the water-level as well as the accuracy of this estimation (Vetra-Carvalho et al., 2020b). This also means that for each landmark that was surveyed, we infer if it is flooded or not if the water-level is above the landmark's

height. More details regarding the four datasets are given in Table 2. A sample image for each location, annotated with the measured landmarks, is given in Fig. 4.





### 4.1.2 Evaluation protocol

As we explained in Section 4.1.1, the images in the datasets used in these experiments are not annotated with binary masks that would allow the pixel-wise evaluation of the semantic segmentation networks. However, the landmark observations (Vetra-Carvalho et al., 2020b) provide us with the binary flooding information for some of the most relevant locations in the image, for our application. In consequence, we believe that the most relevant way to evaluate our approach is to consider it as a binary landmark classification problem, and to use the typical evaluation criteria related to binary classification (e.g., Gu et al., 2018; Bargoti and Underwood, 2017; Salehi et al., 2017). Note that these criteria are also commonly used in hydrology to evaluate the performance of flood modelling methods for flood extent estimation (e.g., Stephens et al., 2014). Therefore, we use the set of criteria presented in Table 3 to describe the performance of our networks, and also provide the corresponding contingency table. The contingency table was computed between the class labels of the landmarks estimated by a human expert examining of the images (Vetra-Carvalho et al., 2020b), and the class labels estimated by our semantic segmentation networks (pixels corresponding to the landmark locations in the images, estimated as flooded or unflooded).

As we explained in Section 3, we consider 8 different network configurations. For each network, we generate the corresponding water segmentation masks of each image of each dataset. We computed the contingency table for the landmark classification for each dataset and each network separately.

### 4.1.3 Landmark classification results

The results are presented in Table 4. For the DIGL, EVES, STRE, and TEWK datasets, our best approaches are the DeepLab networks trained on the LAGO dataset. Indeed, these networks are able to classify the landmarks with balanced accuracy (BA) of $0.95, 0.97, 0.91$ and $0.95$, respectively, and they always obtain good scores for bias, $F$, $F^1$, $F^2$, $F^3$ and $F^4$. When comparing the corresponding bias (Table 4) to the proportion of flooded landmarks (Table 2), we can observe that these best approaches (DeepLab networks trained on the LAGO dataset) tend to estimate slightly more flooded landmarks than what is expected. However, in comparison with the other networks, they tend to show the lowest false alarm rates (F) and have slightly lower performance for hit rates (H). This shows that they are less prone to overprediction than the other networks at the expense of a slightly higher number of false unflooded (B) landmarks predictions.

On average, the DeepLab architecture pre-trained over COCO-stuff obtains better detection performance than the ResNet50-UperNet architecture pre-trained over ADE20k. The only criteria for which ResNet50-UperNet is competitive with DeepLab is the hit rate (H). This means that the networks tend to predict landmarks marked as flooded with an accuracy on par with DeepLab.

While 2STEPS and WHOLE fine-tuning strategies have very similar performance with BA and $F^1$ scores, overall, 2STEPS has lower bias, which makes 2STEPS approaches perform better with $F^2$ and $F^4$ criteria that penalise overprediction, while the WHOLE approaches perform better with $F^3$ and H criteria that penalise underprediction.

The biggest difference can be noticed with the dataset that was used for the fine-tuning: the networks fine-tuned over LAGO have a clear advantage over the ones fine-tuned over WATERDB. This difference is especially noticeable on two out of four





| Name | Equation | Description |
|---|---|---|
| Balanced Accuracy (BA) | $0.5 \times \frac{A}{A+D} + 0.5 \times \frac{B}{B+C}$ | Range: $[0,1]$. Best possible score: 1 |
| Bias | $\frac{A+C}{A+B}$ | Balance between flooded and unflooded landmark estimation. Range: $[0,\infty]$. Best possible score: proportion flooded |
| Hit rate (H) | $\frac{A}{A+D}$ | Fraction of observed flood landmarks correctly predicted. Range: $[0,1]$. Best possible score: 1 |
| False alarm rate (F) | $\frac{C}{B+C}$ | Fraction of observed unflooded landmarks incorrectly predicted. Range: $[0,1]$. Best possible score: 0 |
| Proportion Correct or $F^1$ | $\frac{A+B}{A+B+C+D}$ | Proportion of correctly estimated landmarks. Biased towards the most common category. Range: $[0,1]$. Best possible score: 1 |
| Critical Success Index (CSI) / Threat Score (or $F^2$) | $\frac{A}{A+C+D}$ | Score that ignores the correct estimation of unflooded landmarks. Range: $[0,1]$. Best possible score: 1. |
| $F^3$ | $\frac{A-D}{A+C+D}$ | Penalises estimations biased towards unflooded landmarks. Range: $[-1,1]$. Best possible score: 1 |
| $F^4$ | $\frac{A-C}{A+C+D}$ | Penalises estimations biased towards flooded landmark prediction. Range: $[-1,1]$. Best possible score: 1 |

**Table 3.** Metrics used to evaluate the algorithms performance. A, B, C, D respectively corresponds to True Flooded (landmark flooded predicted as flooded), True Unflooded (landmark unflooded predicted as unflooded), False Flooded (landmark unflooded predicted as flooded), False Unflooded (landmark unflooded predicted as flooded).





datasets: mostly TEWK, but also STRE. For both STRE and TEWK datasets, fine-tuning the networks over WATERDB decreases the capacity of the network to detect the flooded landmarks. When we look at Table 2, we can see that the TEWK dataset contains the largest number of flooded landmarks, and STRE the second largest. Our hypothesis is that since the WATERDB dataset contains a larger proportion of images with small water segments (e.g., fountains, puddles, etc.). The networks fine-tuned over WATERDB have more difficulties generating large water segments that would be necessary for STRE and

TEWK. This hypothesis is supported by the fact that the $F^3$ scores that penalise underprediction are higher for the networks fine-tuned over LAGO.

Given these observations, using the DeepLab network fine-tuned over the LAGO dataset with a 2STEPS strategy is the best configuration to use.

### 4.1.4 Estimating the water-level using the landmark classification

In order to understand the relationship of the results presented in the previous section with water-level estimation, we developed a simple algorithm (**LBWLE**, for Landmark Based Water-Level Estimation) that aims at estimating the water-level by using the landmark classification information. We estimate the water-level height $\hat{w}$ as the average of a lower bound landmark height $h_{lb}$ and an upper bound landmark height $h_{ub}$, that is $\hat{w} = \frac{h_{lb}+h_{ub}}{2}$.

However, while the most obvious approach would be to consider the lower bound $lb$ as the highest flooded landmark, and

the upper bound landmark $ub$ as the lowest unflooded landmark, we need to reconsider: indeed, even if our networks have relatively high classification accuracy, we need to manage the possibility that landmarks with lower heights are estimated as unflooded while landmarks with higher heights might be estimated as flooded. Thus, our LBWLE method uses a slightly different approach:

Let $\hat{F} \in [0,1]^N$ be the estimated flood state of the $N$ landmarks, sorted by increasing order of height $h_i$, and $k$ be the index

of the highest flooded landmark $k = \max\{i | \hat{F}_i = 1, i = 1, \ldots, N\}$. If we define $U = \sum_{i=1}^{k}(1 - \hat{F}_i)$ as the number of unflooded landmarks between 1 and $k$, then the lower bound index $lb$ is defined as $lb = \lceil k - U\frac{U}{k} \rceil$, and the upper bound $ub$ is defined as $ub = lb + 1$. With this algorithm, the idea is to first consider the lower bound index $lb$ as the index of the highest landmark estimated as flooded, but to switch to lower landmark indices, depending on the percentage of unflooded landmarks between 1 and $k$. An example for the choice of the lower bound index using LBWLE is given in Fig. 5.

The estimated river level height $\hat{w}$ will then be estimated as the average between those the heights of the landmarks defined as the lower and upper bounds $\hat{w} = \frac{h_{lb}+h_{ub}}{2}$. If no landmark is estimated as flooded, then the water-level is set to $\hat{w} = h_1$ (the lowest water-level measured), and if all the landmarks are estimated as flooded, then the water-level is set to $\hat{w} = h_N$ (the highest water-level measured). Note that the accuracy of LBWLE is dependent on the annotated landmarks as it can only estimate the water level as the average height of two landmark heights.

We show the results of this LBWLE estimation method applied on our best performing network (DeepLab-LAGO-2STEPS) in Fig. 6. For Diglis Lock, Evesham and Strensham, Fig. 6 shows that, for the evaluated two-week flood event period, our method was able to give a good approximation of the manually estimated water-level. Indeed, we can see that LBWLE's estimation and the water-level estimated by a human observer almost always have the same landmarks as lower and upper





| | Contingency table | | | | Metrics | | | | | | | |
|---|---|---|---|---|---|---|---|---|---|---|---|---|
| | A | B | C | D | BA | bias | $F^1$ | $F^2$ | $F^3$ | $F^4$ | H | F |
| **Diglis Lock (DIGL)** | | | | | | | | | | | | |
| RU-LAGO-WHOLE | 231 | 686 | 63 | 7 | 0.94 | 0.32 | 0.93 | 0.77 | 0.74 | 0.56 | 0.97 | 0.08 |
| RU-LAGO-2STEPS | 229 | 688 | 61 | 9 | 0.94 | 0.32 | 0.93 | 0.77 | 0.74 | 0.56 | 0.96 | 0.08 |
| RU-WATERDB-WHOLE | **238** | 657 | 92 | **0** | 0.94 | 0.37 | 0.91 | 0.72 | 0.72 | 0.44 | **1.00** | 0.12 |
| RU-WATERDB-2STEPS | 234 | 656 | 93 | 4 | 0.93 | 0.37 | 0.90 | 0.71 | 0.69 | 0.43 | 0.98 | 0.12 |
| DeepLab-LAGO-WHOLE | 230 | **704** | **45** | 8 | **0.95** | **0.29** | **0.95** | **0.81** | **0.78** | **0.65** | 0.97 | **0.06** |
| DeepLab-LAGO-2STEPS | 229 | 695 | 54 | 9 | **0.95** | 0.31 | 0.94 | 0.78 | 0.75 | 0.60 | 0.96 | 0.07 |
| DeepLab-WATERDB-WHOLE | 231 | 673 | 76 | 7 | 0.93 | 0.34 | 0.92 | 0.74 | 0.71 | 0.49 | 0.97 | 0.10 |
| DeepLab-WATERDB-2STEPS | 235 | 688 | 61 | 3 | **0.95** | 0.32 | 0.94 | 0.79 | **0.78** | 0.58 | 0.99 | 0.08 |
| **Evesham (EVES)** | | | | | | | | | | | | |
| RU-LAGO-WHOLE | 495 | 1145 | 58 | 44 | 0.94 | 0.34 | 0.94 | 0.83 | 0.76 | 0.73 | 0.92 | 0.05 |
| RU-LAGO-2STEPS | 494 | 1163 | 40 | 45 | 0.94 | **0.32** | 0.95 | 0.85 | 0.78 | 0.78 | 0.92 | 0.03 |
| RU-WATERDB-WHOLE | 505 | 1103 | 100 | 34 | 0.93 | 0.38 | 0.92 | 0.79 | 0.74 | 0.63 | 0.94 | 0.08 |
| RU-WATERDB-2STEPS | 454 | 1166 | 37 | 85 | 0.91 | 0.30 | 0.93 | 0.79 | 0.64 | 0.72 | 0.84 | 0.03 |
| DeepLab-LAGO-WHOLE | **521** | 1168 | 35 | **18** | **0.97** | 0.33 | **0.97** | **0.91** | **0.88** | 0.85 | **0.97** | 0.03 |
| DeepLab-LAGO-2STEPS | 516 | **1176** | **27** | 23 | **0.97** | **0.32** | **0.97** | **0.91** | 0.87 | **0.86** | 0.96 | **0.02** |
| DeepLab-WATERDB-WHOLE | 518 | 1090 | 113 | 21 | 0.93 | 0.39 | 0.92 | 0.79 | 0.76 | 0.62 | 0.96 | 0.09 |
| DeepLab-WATERDB-2STEPS | 490 | 1150 | 53 | 49 | 0.93 | 0.33 | 0.94 | 0.83 | 0.74 | 0.74 | 0.91 | 0.04 |
| **Strensham Lock (STRE)** | | | | | | | | | | | | |
| RU-LAGO-WHOLE | 1194 | 1866 | 306 | 90 | 0.89 | 0.49 | 0.89 | 0.75 | 0.69 | 0.56 | 0.93 | 0.14 |
| RU-LAGO-2STEPS | 1200 | 1882 | 290 | 84 | 0.90 | 0.48 | 0.89 | 0.76 | 0.71 | 0.58 | 0.93 | 0.13 |
| RU-WATERDB-WHOLE | **1260** | 1609 | 563 | **24** | 0.86 | 0.64 | 0.83 | 0.68 | 0.67 | 0.38 | **0.98** | 0.26 |
| RU-WATERDB-2STEPS | 1230 | 1658 | 514 | 54 | 0.86 | 0.60 | 0.84 | 0.68 | 0.65 | 0.40 | 0.96 | 0.24 |
| DeepLab-LAGO-WHOLE | 1200 | 1905 | 267 | 84 | **0.91** | 0.47 | **0.90** | 0.77 | **0.72** | 0.60 | 0.93 | 0.12 |
| DeepLab-LAGO-2STEPS | 1191 | **1923** | **249** | 93 | **0.91** | **0.46** | **0.90** | **0.78** | **0.72** | **0.61** | 0.93 | **0.11** |
| DeepLab-WATERDB-WHOLE | 1167 | 1866 | 306 | 117 | 0.88 | 0.49 | 0.88 | 0.73 | 0.66 | 0.54 | 0.91 | 0.14 |
| DeepLab-WATERDB-2STEPS | 1148 | 1869 | 303 | 136 | 0.88 | 0.48 | 0.87 | 0.72 | 0.64 | 0.53 | 0.89 | 0.14 |
| **Tewkesbury Marina (TEWK)** | | | | | | | | | | | | |
| RU-LAGO-WHOLE | 221 | 289 | 22 | 20 | 0.92 | 0.48 | 0.92 | 0.84 | 0.76 | 0.76 | 0.92 | 0.07 |
| RU-LAGO-2STEPS | 225 | **299** | **12** | 16 | **0.95** | 0.45 | **0.95** | **0.89** | 0.83 | **0.84** | 0.93 | **0.04** |
| RU-WATERDB-WHOLE | 214 | 247 | 64 | 27 | 0.84 | 0.60 | 0.84 | 0.70 | 0.61 | 0.49 | 0.89 | 0.21 |
| RU-WATERDB-2STEPS | 172 | 282 | 29 | 69 | 0.81 | **0.44** | 0.82 | 0.64 | 0.38 | 0.53 | 0.71 | 0.09 |
| DeepLab-LAGO-WHOLE | **233** | 288 | 23 | **8** | **0.95** | 0.49 | 0.94 | 0.88 | **0.85** | 0.80 | **0.97** | 0.07 |
| DeepLab-LAGO-2STEPS | 229 | 295 | 16 | 12 | **0.95** | 0.47 | **0.95** | **0.89** | 0.84 | 0.83 | 0.95 | 0.05 |
| DeepLab-WATERDB-WHOLE | 144 | 297 | 14 | 97 | 0.78 | 0.36 | 0.80 | 0.56 | 0.18 | 0.51 | 0.60 | 0.05 |
| DeepLab-WATERDB-2STEPS | 192 | 288 | 23 | 49 | 0.86 | 0.45 | 0.87 | 0.73 | 0.54 | 0.64 | 0.80 | 0.07 |

**Table 4.** Landmark detection results (for the metric meanings, see Table 3). For each location and each metric, the best network results are in bold. RU stands for the ResNet50-UperNet network.





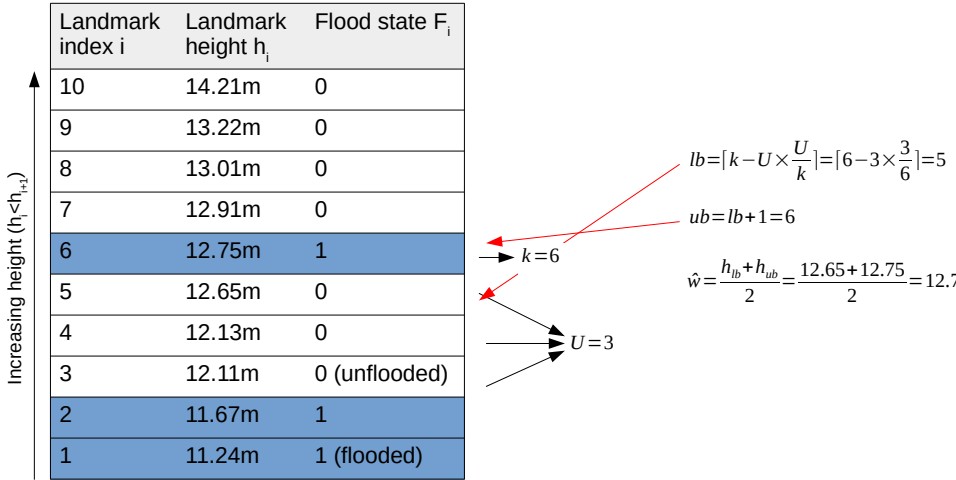

**Figure 5.** Example of application of the LBWLE algorithm. Its principle is that if some of the highest landmarks are estimated as flooded, but some lower height landmarks are estimated as unflooded, then the true water-level is likely lower than the height of the highest landmark estimated as flooded.

bounds, which is as close as LBWLE's performance can achieve as it is limited by the heights of the landmarks that were

measured during the ground survey (the dotted lines in Fig. 6). We notice that only a few estimation mistakes were made on the Tewkesbury Marina dataset: out of 138 images, only 5 estimation mistakes were made. Those mistakes were due to a landmark that was annotated on a platform close to a building. In this case, the networks stretched the unflooded segmentation area (related to the building) to the landmark location.

### 4.2   Performance evaluation for year long water-level analysis

**4.2.1   Year-long river-camera images datasets**

For this experiment, we considered the same camera locations as those used for the first experiment presented in Section 4.1.1. However, we used a different, one year period from 1 June 2019 to 31 May 2020. According to a government report (Finlay, 2020), three major flood events occurred during this period. The first one, in November, was due to heavy rainfall at the start of the month (7-8 November), followed by additional heavy rainfall between the 13 and 15 November. The second major event

happened in the second half of December, with heavy rain pushing across the southern parts of England, and lasting until the New Year 2020. Finally, the storms Ciara, Dennis and Jorge swept across the U.K from 9 February 2020 to the early days of March. Additionally, heavy rainfall occurred between 10-12 June 2019.

    For Diglis Lock, Evesham, Strensham Lock and Tewkesbury Marina, we have 3081, 3012, 3067 and 3147 images, respectively. The difference in the number of images is due to minor technical camera problems making some images unavailable.



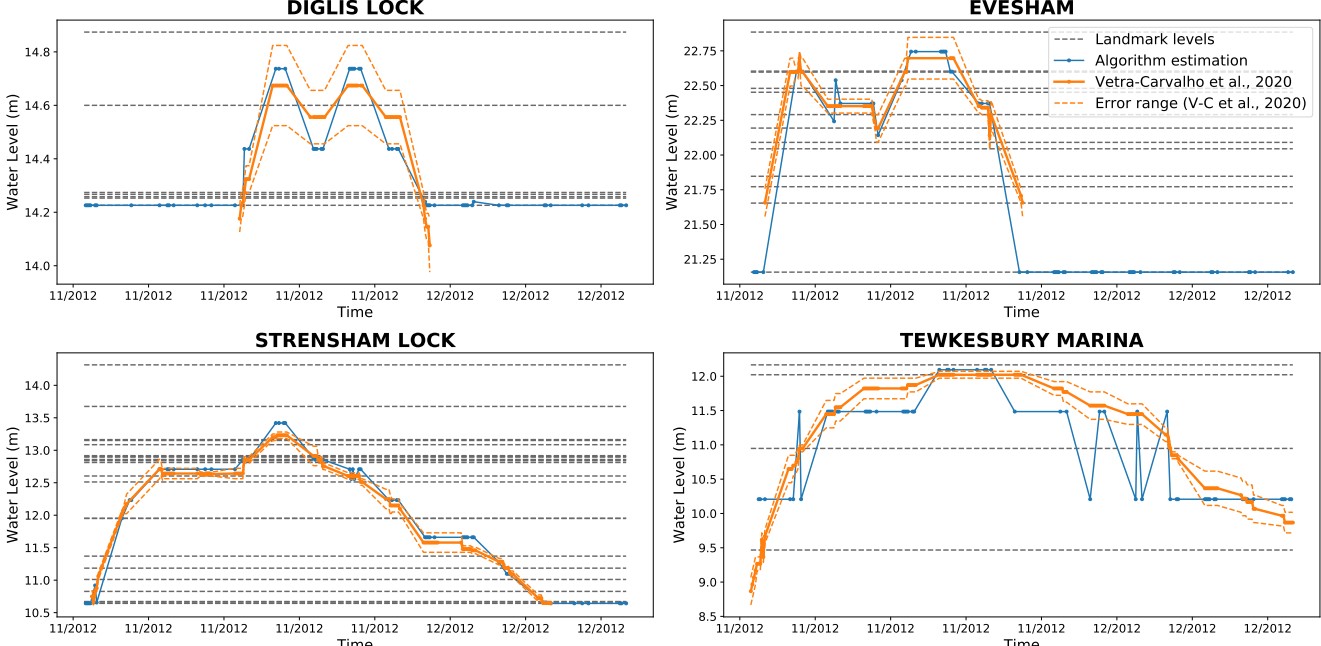

**Figure 6.** Comparison of our water-level estimation method using the DeepLab-LAGO-2STEPS network (in blue) using the landmarks with the ground truth water-levels directly extracted from the images (Vetra-Carvalho et al., 2020b) (in orange). The horizontal dashed lines correspond to the heights of the landmarks ground-surveyed on these locations (see Section 4.1.1) that can be used as lower and upper bounds by our water-level estimation algorithm LBWLE (see Section 4.1.4). Note that the water-level estimation performed by manual examination of the images (Vetra-Carvalho et al., 2020b) was not always available outside of the flood event itself (Diglis Lock, Evesham and Strensham).

The Diglis Lock and Tewkesbury Marina camera mounting positions, orientation and fields-of-view were changed in 2016 (Vetra-Carvalho et al., 2020b), so they are different from the first experiment (see Fig. 4). The new fields-of-view are presented in Fig. 7. The original RGB image size for these datasets is $640 \times 480$, which is a lower image resolution than in the first experiment. As the Diglis Lock and Tewkesbury Marina camera locations were changed, the corresponding landmarks used in the first experiment can not be considered for this experiment.

The water-levels were not manually annotated on these year-long datasets. In order to evaluate the relevance of our algorithm on these datasets, we used the water-level information coming from nearby river-gauges available through the UK's Environment Agency open data API (Environment Agency (2020)). The water-level information from the river-gauges is not expected to reflect the exact situation observed at the camera location, but the water-levels should be highly correlated. The locations of the gauges are given in Table 5 of Vetra-Carvalho et al. (2020b). The distance from the camera to their nearest river gauge ranges from 51m to 1823m.





**Diglis Lock**

**Tewkesbury Marina**

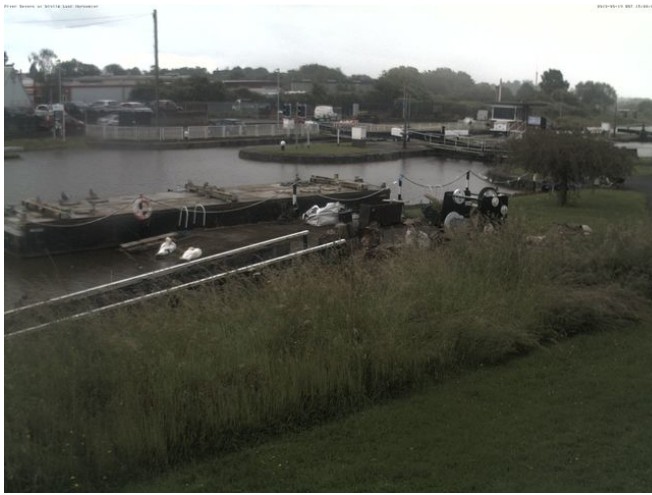
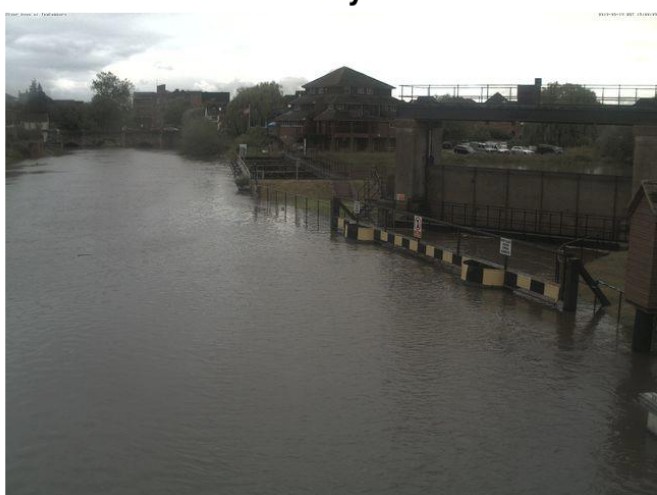

**Figure 7.** Fields-of-view from Diglis Lock and Tewkesbury Marina cameras for the period 2019-2020.

### 4.2.2 Evaluation protocol

Given that we are not able to use the landmarks from the ground-survey on two of the four cameras that were used in the first experiment and independent water-level information for validation is from nearby rather than co-located river-gauges, we cannot use the same protocol that we developed for the first experiment. Hence, after applying our water semantic segmentation network algorithms on the images, we designed two experiments:

1. **Landmark-based water level estimation analysis.** For the images from the two locations for which the annotated landmark locations are still valid (Evesham and Strensham Lock), we measured the correlation between the water-levels measurements from the nearest river-gauges, and the water-levels estimated by applying the LBWLE algorithm that we presented in Section 4.1.4 on the images segmented by our water semantic segmentation networks. The correlation between $N$ estimations of water-levels, with $w$ being the LBWLE estimation, and $g$ being the corresponding nearest river-gauge water-level measurement is computed using PCC, Pearson's correlation coefficient (Freedman et al., 2007), as defined in Eq. (1).

$$\rho = \frac{\sum_i^N (w_i - \bar{w})(g_i - \bar{g})}{\sqrt{\sum_i^N (w_i - \bar{w})^2}\sqrt{\sum_i^N (g_i - \bar{g})^2}} \tag{1}$$

where $\bar{w} = \frac{1}{N}\sum_i^N w_i$ and $\bar{g} = \frac{1}{N}\sum_i^N g_i$.

2. **Full image SOFI index analysis.** For each of the four locations, we computed the Pearson's correlation coefficient between the water-level measurements obtained from the nearest river-gauge, and the SOFI index (Moy de Vitry et al.,





|  | Evesham Lock | Strensham Lock |
|---|---|---|
|  | (EVES) | (STRE) |
| RU-LAGO-WHOLE | 0.69 | 0.89 |
| RU-LAGO-2STEPS | 0.7 | 0.86 |
| RU-WATERDB-WHOLE | 0.71 | 0.88 |
| RU-WATERDB-2STEPS | 0.77 | 0.91 |
| DeepLab-LAGO-WHOLE | 0.65 | 0.91 |
| DeepLab-LAGO-2STEPS | 0.71 | 0.91 |
| DeepLab-WATERDB-WHOLE | 0.77 | 0.92 |
| DeepLab-WATERDB-2STEPS | 0.72 | 0.92 |

**Table 5.** Pearson's Correlation Coefficients computed between the landmark-based water-level estimation and the water-levels from the nearest river-gauges on Evesham and Strensham Lock dataset.

2019) computed on the segmented images. The SOFI index corresponds to the percentage of water pixels in a segmented image (the number of pixels estimated as flooded divided by the total number of pixels in the image).

### 4.2.3 Landmark-based water level estimation analysis

For the images from the two locations for which the annotated landmark locations are still valid (EVES and STRE), Table 5 shows the correlation between the nearest river-gauge water-level measurements and our water-level estimation using the LB-WLE algorithm presented in Section 4.1.4. For these images, we find that the networks that were trained on WATERDB obtain among the highest correlations. This is especially the case for the DeepLab networks. The DeepLab networks obtain higher correlations than the ResNet50-UperNet networks. The 2STEPS fine-tuning approach has a slight advantage over WHOLE

fine-tuning. However, these differences stay relatively small as the camera location has a higher influence on the correlation. Thus, dataset-specific parameters have a significant influence over the results: Table 4 (computed for the first experiment) shows that the Evesham landmarks get generally better detection results than the Strensham Lock landmarks, so it seems that is not related to a location being more easily segmented than another.

If we consider the corresponding time-evolution of the water-levels in Fig. 8, we can explain the highest correlations at

Strensham Lock by the fact that the Evesham landmark heights do not allow tracking of the typical lower water levels when the river is in-bank, while the landmarks at Strensham Lock allow better tracking of the water-level at lower heights.

In addition, as the river-gauge used for Strensham Lock (Eckington Sluice) is 51m away from the camera whereas the nearest river-gauge to the Evesham camera is 1823m away (Vetra-Carvalho et al., 2020b), it could be expected that the water levels extracted from the nearest river-gauge at Strensham depict a more representative evolution of the water levels at Strensham

Lock than the river-gauge used for Evesham. Also, note that at Strensham, the lock can affect the water-level.



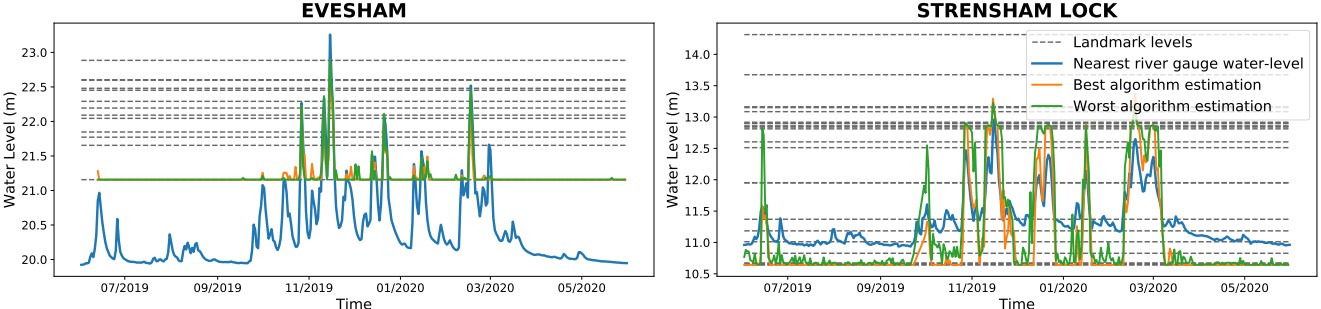

**Figure 8.** Evesham and Strensham Lock year-long water-levels, measured with using landmark annotations, in comparison with water-levels from nearby river-gauges. The best networks are DeepLab-WATERDB-WHOLE for Evesham and Strensham. The worst networks are RU-LAGO-WHOLE for Evesham and RU-LAGO-2STEPS for Strensham.

### 4.2.4 Full image SOFI index analysis

As this experiment does not require the landmark information, for each of the four locations and each network , the correlation of the SOFI index (percentage of pixels estimated as water pixels in the image) with the water-levels from the nearest river-gauges is shown in Table 6. In Fig. 9, we show the corresponding standardised water-levels and the standardised SOFI indexes

with the highest and lowest correlation with the water-level, produced with the corresponding networks shown in Table 6. In this work, we use the term standardisation to describe the process of putting different variables on the same scale. In order to standardise the observed value $X_i$ of a variable $X$, we consider the difference of this observed value $X_i$ with the mean (time-average) of the variable $\bar{X}$ and divide this difference with the standard deviation of the variable $\sigma(X)$. So, if $X_i^S$ is the standardised observed value corresponding to $X_i$, then $X_i^S = \frac{X_i - \bar{X}}{\sigma(X)}$.

In Table 6, we see that the correlations of the 8 networks with the river-gauge water-levels are relatively similar and that the difference in-between datasets is much more obvious: the lowest correlation on Strensham is higher than the highest correlation obtained on Evesham. The lowest correlation obtained on Evesham is higher than the highest correlation obtained on Diglis, and the lowest correlation on Diglis is higher than the highest correlation on Tewkesbury. The correlation results are especially low for the Tewkesbury Marina location, where some correlations are close to zero/negative. For Strensham and Evesham, we

can also notice that the correlations using the SOFI index are higher than the correlations obtained when using the landmark information (see Table 5).

We can explain the higher correlations in Table 6 in comparison with Table 5 by examining the evolution of the water levels in Fig. 9. Fig. 9 shows that the SOFI index allows the algorithms to provide a better estimate of the water-level when the river is in-bank than the landmark-based estimation. However, we can notice that the estimates, when the water-levels are low, stay

fairly approximate and subject to small perturbations. Indeed, at low water-level, we can see changes in the SOFI indexes that are not correlated with any particular event. When we analysed the results on the Tewkesbury Marina dataset, where we can see that this phenomenon is the strongest, our visual inspection of the water segmentation results showed us that the segmentation





|  | Diglis Lock | Evesham | Strensham Lock | Tewkesbury Marina |
|---|---|---|---|---|
|  | DIGL | EVES | STRE | TEWK |
| RU-LAGO-WHOLE | 0.69 | 0.89 | 0.94 | -0.08 |
| RU-LAGO-2STEPS | 0.66 | 0.90 | 0.93 | -0.03 |
| RU-WATERDB-WHOLE | 0.73 | 0.91 | 0.94 | 0.08 |
| RU-WATERDB-2STEPS | 0.71 | 0.80 | 0.93 | 0.19 |
| DeepLab-LAGO-WHOLE | 0.58 | 0.90 | 0.93 | 0.16 |
| DeepLab-LAGO-2STEPS | 0.60 | 0.91 | 0.93 | 0.09 |
| DeepLab-WATERDB-WHOLE | 0.72 | 0.87 | 0.93 | 0.16 |
| DeepLab-WATERDB-2STEPS | 0.67 | 0.86 | 0.93 | 0.07 |

**Table 6.** Pearson's correlation coefficients computed between the SOFI index and the water-levels obtained from the nearest river-gauges.

networks worked correctly. However, we noticed that due to the new field-of-view of the camera, and the configuration of the location, floods were not heavily increasing the number of water pixels in the image, and thus did not result in large increase

of the SOFI index. The occlusion of some water segments in the image due to passage or mooring of boats could have a significant influence on the SOFI index results, and thus explains the uncorrelated SOFI index changes for this dataset. In all the locations, we can also observe smaller, noisy, perturbations of the SOFI index when the water-level is low and steady. These perturbations are due to various, smaller-scale, problems: occlusions by boats or changes in the lock configuration (there is a cable ferry at the Evesham location, and the other locations are all locks), small segmentation errors or approximations from

our segmentation algorithm. Besides, it is also likely that depending on the site configuration (e.g, the slope of the area close to the river) and the field of view of the camera, water-level changes can have varied impacts on the SOFI index.

### 4.2.5 Windowed image SOFI index analysis

Given the remarks made in the previous section (Section 4.2.4) regarding the impact of the field of view of a camera and the possible occlusion of some water segments in the image, we developed a technique to compute the SOFI indexes over smaller

regions (windows) within the image, where the SOFI index could give a more accurate description of the water-level evolution.

For this experiment, we partitioned the images into a $4 \times 4$ grid of windows of equivalent size (image height/4, image width/4), and looked for the window with the SOFI index which was the most correlated with the water-level obtained from the nearest river-gauge. If the correlation obtained using the SOFI index of the entire image was higher, then we kept the SOFI index of the entire image. In order to avoid overfitting our datasets during the selection of this window, the choice was made

using a validation dataset consisting of the river-camera images and river-gauge levels dating from 2018 (every available image between 1 January 2018 and 31 December 2018).

The results of this last experiment are shown in Table 7 and Fig. 10. At Diglis Lock, Evesham and Tewkesbury Marina, we can observe higher correlations with the nearest river-gauges. This experiment did not change the results for Strensham Lock as the SOFI index computed for the entire image was selected during validation.





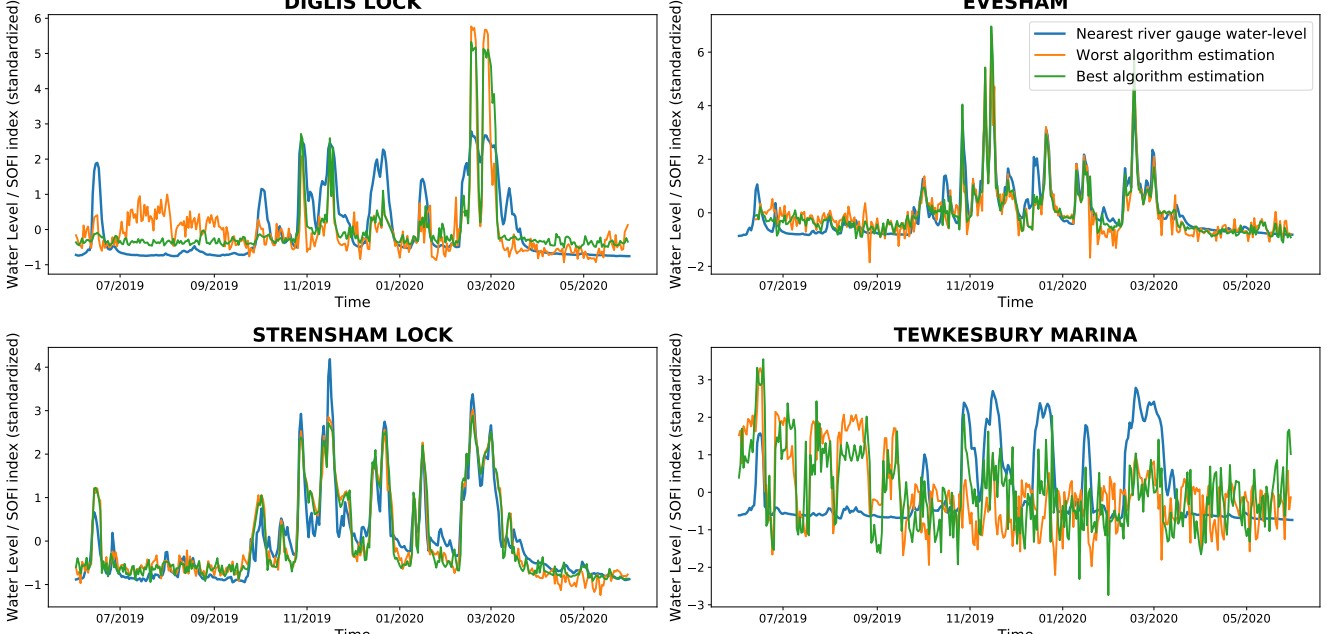

**Figure 9.** standardised SOFI indexes in comparison with standardised water-levels from nearby river-gauges. For each location, the best and worst algorithms can be found in Table 6.

For all the datasets, the standardised SOFI index computed over the water segmentation of the window is able to accurately fit the standardised evolution of the water-level obtained from the nearby river-gauges, both at low and high water-levels. As with the previous experiments, we do not see the clear dominance of a particular CNN, fine-tuning dataset or methodology. This is highlighted in Fig. 10, where we can see that the best and worst algorithms have very similar behaviour. This could be explained by the fact that the choice of the best window is also conditioned by the relative facility for the networks to segment

the water inside it. We can also observe that there is a reduction in noise for low water-levels compared with Fig. 9: the choice of window has reduced the impact of occlusions, and the noise level is also likely influenced by the performance of the network on the area.

In Fig. 11, we show the best windows selected during the validation process by the 8 different networks. We can see that the same window location is selected for each of the networks for three out of four locations. For Diglis, the only exception,

we can observe that both windows offer similar perspectives in terms of water/land surfaces. For Strensham, keeping the SOFI index computed over the entire image gives the best correlation. If such a window location had to be chosen in a different context without a nearby gauge for comparison, a possible heuristic could be to choose a location with a roughly equal areas of land/water surfaces where the river level can increase progressively over the land surface (land surfaces with small slopes are preferred).



|  | Diglis Lock | Evesham | Strensham Lock | Tewkesbury Marina |
|---|---|---|---|---|
|  | DIGL | EVES | STRE | TEWK |
| RU-LAGO-WHOLE | 0.90 | 0.97 | 0.94 | 0.96 |
| RU-LAGO-2STEPS | 0.90 | 0.97 | 0.93 | 0.96 |
| RU-WATERDB-WHOLE | 0.90 | 0.97 | 0.94 | 0.94 |
| RU-WATERDB-2STEPS | 0.92 | 0.95 | 0.93 | 0.95 |
| DeepLab-LAGO-WHOLE | 0.90 | 0.97 | 0.93 | 0.96 |
| DeepLab-LAGO-2STEPS | 0.90 | 0.98 | 0.93 | 0.96 |
| DeepLab-WATERDB-WHOLE | 0.94 | 0.94 | 0.93 | 0.97 |
| DeepLab-WATERDB-2STEPS | 0.94 | 0.96 | 0.93 | 0.95 |

**Table 7.** Pearson's correlation coefficients computed between the SOFI indexes of the best window from the $4 \times 4$ grid and the water-levels obtained from the nearest river-gauges.

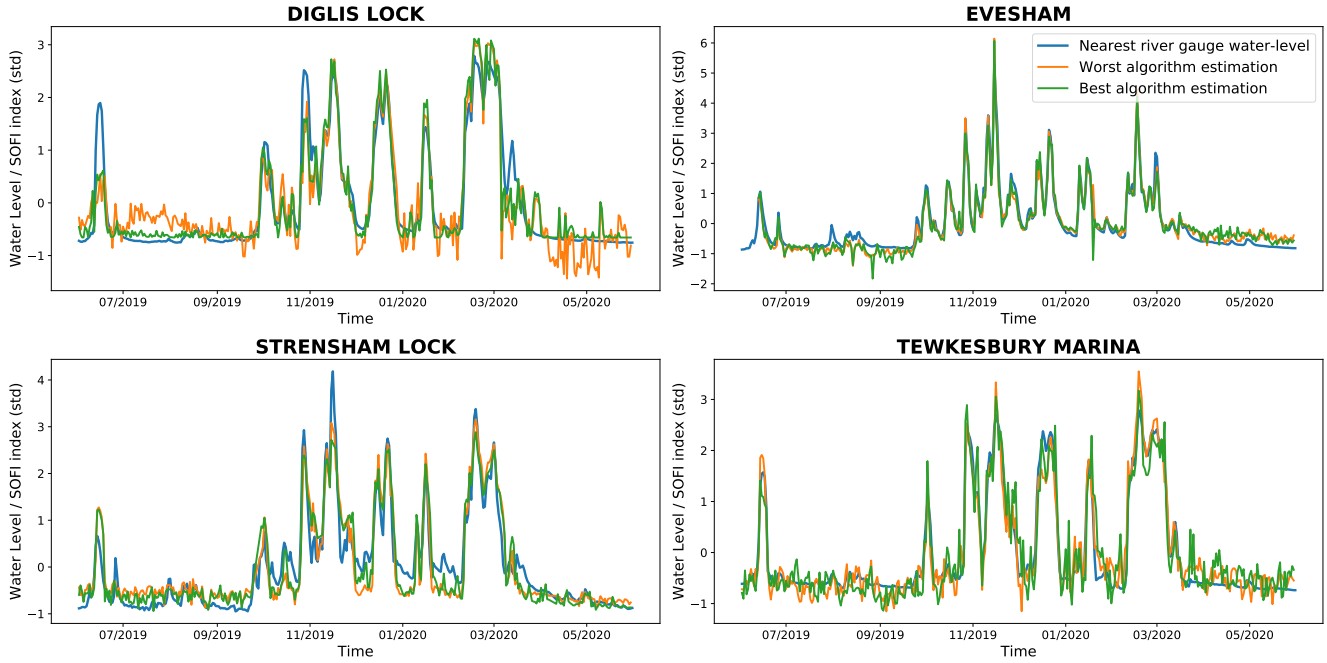

**Figure 10.** standardised SOFI indexes of the best window from the $4 \times 4$ grid in comparison with standardised water-levels from nearby river-gauges.





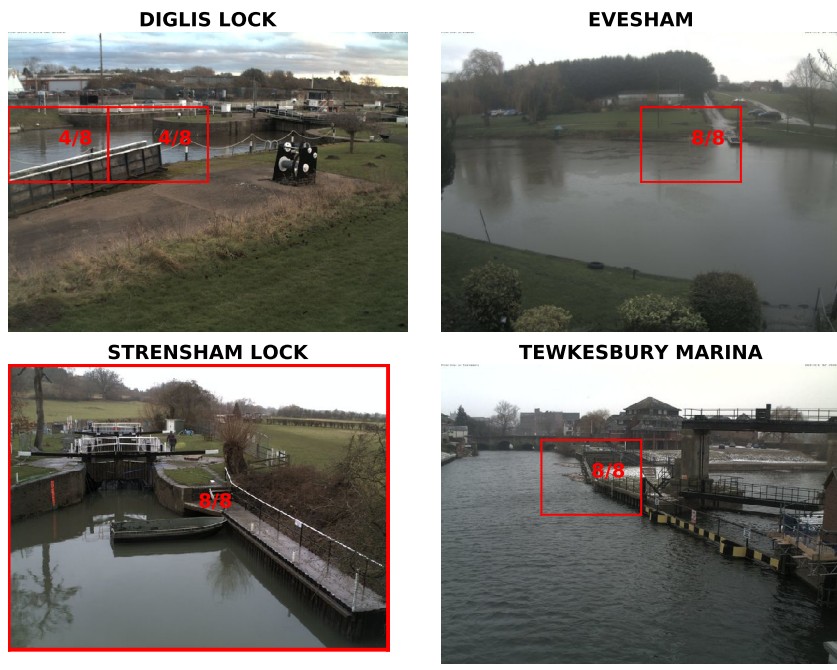

**Figure 11.** Windows of the $4 \times 4$ grid where the segmentation gives the best correlation with the water-level, for at least one of the $8$ networks considered. The fractions correspond to the proportion of networks that selected the corresponding window as the one giving the best correlation

## 5 Conclusions


In this paper, we addressed the problem of water segmentation using river cameras images to automate the process of water-level estimation. We tackled the problem of water segmentation by applying TL techniques to deep semantic segmentation networks trained on large datasets of natural images.

With our first experiment regarding the classification of landmarks annotated with water-level information on small two-

week datasets, we showed that our best water segmentation networks were able to reach balanced accuracy greater than $91\%$ for each of the studied locations, which both proved the good segmentation performance of our algorithm and showed its potential in the context of flood extent analysis studies.

We then developed an algorithm, LBWLE, in order to directly estimate the water-level from the classified landmarks, and showed that we were able to estimate the water-level with the maximum accuracy this algorithm could reach, as this approach

is inherently limited by the heights of the landmarks used for the study. Given a camera location and a detailed ground survey in the field-of-view of the camera, this approach can provide an accurate estimation of the water level at the camera location.

With our second experiment, we used much larger, year-long datasets of images with no water-level annotations available. We used available water-levels from nearby river-gauges as validation data, and showed that the water-levels estimated using

the LBWLE approach could also be used in this context. Indeed, with this approach, we were able to measure the water-levels for the three major floods that happened during the year.

We then investigated the use of the SOFI index (Moy de Vitry et al., 2019) (percentage of water pixels in the image) applied on the entire image to show our results were strongly correlated with the water-level from the nearby river-gauges, showing that it was possible to use the SOFI index to track flood events, and have a better tracking of lower flows while the river is still in-bank than when using LBWLE. However, for one location, occlusions occurring in the field-of-view of the camera impacted

the results.

Finally, we developed a simple approach that computes the SOFI index on a specific window (sub-region) of the image. This window is selected through a simple validation procedure using older images and water-levels from the same locations. With this approach, we were able to accurately track large flood events as well as smaller changes while the river is still in-bank on all of our datasets. While this approach is the most accurate that we developed during this study, the choice of the window

relies on relatively close river gauges. However, we suggested some straightforward guidelines in order to help the potential user to chose the window if nearby gauges are not available.

The algorithms and experiments presented in this study show a great potential of transfer learning and semantic segmentation networks for the automation of the water-level estimations. These methods could drastically reduce the costs and workloads related to the evaluation of water-levels, which is necessary for many applications, including the understanding of the ever

increasing number of flood events.

*Code and data availability.* The images and annotations used in Section 4.1 are available in Vetra-Carvalho et al. (2020a). The networks used for our experiments and the images used in Section 4.2 can be found in Vandaele et al. (2020b). The river gauge data can be found on the Environment Agency website (https://environment.data.gov.uk/)

.

*Author contributions.* R.V implemented the methods and experiments, and was the main writer of the manuscript. S.L.D. was the project principal investigator, obtained the funding for the work and set the overarching goals for the project. V.O was the main advisor for the deep-learning related aspects of the study. S.L.D and V.O both contributed to the improvement of the manuscript.

*Competing interests.* The authors have no competing interests.

*Acknowledgements.* This work was funded by the Data Assimilation for the REsilient city (DARE) project, an EPSRC Senior Fellowship
in Digital Technology for Living with Environmental Change (EPSRC EP/P002331/1). The authors would like thank Glyn Howells from



Farson Digital Ltd for granting access to camera images. The authors would also like to thank David Mason, University of Reading, for useful discussions.





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
