# Peer review of "Deep learning for automated river-level monitoring through river camera images: an approach based on water segmentation and transfer learning"

_Hydrology and Earth System Sciences, 2021_

## Referee Comment (RC2)

Comments to the Author:

Thanks for submitting the interesting manuscript, and I really appreciate your attempt to use deep learning technique to solve the problems in the field of hydrology. Before your revision, I would like you to consider my following suggestions:

1. emphasize the scientific significance of your research.
The novelty of the paper hasn't been clearly proposed. The main contents of your work can be classified into two parts: firstly, water segmentation based on transfer learning approaches; secondly, the development of corresponding algorithms to monitor the water level with or without the help of landmarks. However, neither the scientific significance of the contents were fully reflected in the paper, especially in introduction and title.

If you want to address the significance of transfer learning, some extra experiments will be necessary. Not only the comparison between different transfer learning approaches should be carried on, but the advantages of transfer learning over directly training models should be clarified with data through series of experiments. I suggest that you design more detailed experiments to explore the relationships between the number of training data and the superiority of transfer learning approaches. This will also help to directly explain why you use transfer learning.

LBWLE is not important or novel as expected. It is highly influenced by the number and quality of the landmarks, so you should focus more on how to increase the accuracy of semantic segmentation model itself as suggested previously. Though the significance of the second point may be smaller, yet if you want to emphasize the second point, there should be more discussions on LBWLE algorithm. The advantage of LBWLE over previous methods (e.g. SOFI) and the necessities of the LBWLE were not fully discussed in the paper. Add some reference on the advantages and disadvantages of the previous methods to show the novelty of your method.

Accordingly, the title also has to be changed to put emphasis on transfer learning, and the introduction need to be restructured. The introduction is suggested to be organized to present more about the development of water semantic segmentation methods and the corresponding water level indexes rather than the history of water level monitoring approaches.

2. Suggestions on language and writing style.
As for language, avoid using too much first person expression like "we". As for writing style, the paper is organized more like a paper published in computer science journals, e.g., separating the introduction and background. adjust your paper structure referred to other papers in HESS if you still would like to publish the paper in a hydrological journal.

3.  Check your reference.

Please make sure every paper you refer to can support your ideas. E.g. in line 26: "The network of river gauging stations is declining globally (Vorosmartyetal., 2001)" , the expression is doubted, if it is the idea in the paper? In line 119, the originality of Moy de Vitry etal.(2019) lied on two parts, the development of a water segmentation model and the proposal of SOFI, the expression in your paper (the biggest originality...) was not rigorous.

4.  Give an explanation when computer science terms first appear, like "fine-tune".

5.  The two transfer learning approaches cannot be fully understand only from the present description, it is recommended to use figure to illustrate it. And comment on what's the essential difference between the two transfer learning approaches.

6.  The structure of the original semantic segmentation deep learning model would be altered when be transferred, e.g., adding an fully-connected layer to adjust the model task from multi-class to two-class. It is a necessary step but not mentioned in your paper.

7.  A lot of metrics for evaluating the results has been listed in Table 3. Are they all useful for the evaluation? Only remain the metrics helpful for the following discussion and explain what the metrics represent before discussing the results.

8.  If the LBWLE can only capture the flood rather than drought (Figure 8)? If so, this will limit the value of the algorithm in hydrological problems such as hydrological model calibration.

9. Illustrate what's your purpose on setting up two experiments (two-week and year-long) ? Do they correspond to different application scenes respectively? What substantial conclusions can be reached through the comparison of two experiments, why not only use the year-long series?

In conclusion, though your research is really interesting, I recommend you to supplement the suggested experiments and reorganize the structure. I'm looking forward to your reply to my questions.

---

## Author Comment (AC2)

**HESS-2021-20**
*Vandaele et al.* **Deep learning for the estimation of water-levels using river cameras**
**Response to Reviewer #2**

We would like to thank the reviewer for their comments and suggestions. We respond to these comments  below. For clarity, we have introduced some extra numbering (1a, 1b, 1c etc.) to address the separate points made in each comment.

As part of our response to the comments,  we propose to revise the organization of our manuscript. The new proposed *table of contents* is located at the end of this document. If our manuscript is accepted for revision, in addition to the changes proposed below, we intend to update parts of the text to better fit this new organization (section introductions for example). for the sake of clarity, we preferred to keep these changes out of this answer.

**Comment #1a: Emphasize the scientific significance of your research, including the transfer learning aspects**

In order to highlight the novelty of our manuscript and answer the reviewer's comment, if our manuscript is accepted for revision, we would like to make the following changes:

(1) Modify the end of the introduction (starting line 58, ending line 69) to give a better introduction to transfer learning and our previous paper:

*"Over the last decade, transfer learning (TL) techniques have become a common tool to try to overcome the lack of available data (Reyes et al, 2015; Sabbatelli et al, 2018). The aim of these techniques is to repurpose efficient machine learning models trained on large annotated datasets of images to new related tasks where the availability of annotated datasets is much more limited (see Section 2 for more details). Vandaele et al (2020) successfully analyzed a set of TL approaches for improving the performance of deep water segmentation networks. In this paper, we build on the work of Vandaele et al. (2020) and study the performance of these water segmentation networks for the automation of river-level estimation from river-camera images, in the context of flood-related studies. In particular, we carry out novel experiments realized with new river-camera datasets and metadata, that consider the use of several methods to extract quantitative water-level observations from the segmented river-camera images."*

(2) Create a new Section 2 that would be our Methodology section and would encompass:
  ● the former Section 2 introducing semantic segmentation, deep learning and transfer learning
  ● the former Section 3 presenting the application of  transfer learning
  ● the presentation of SOFI and the new LBWLE method moved from former sections 4.1.4 and 4.2.2

We think that this new organization will help the reader to understand our research more clearly.

**Comment #1b Importance and novelty of LBWLE**

Our goal with LBWLE was to propose a way to provide quantitative water-level observations in accepted units (m), using the landmark-annotated dataset at our disposal.

If our manuscript is accepted for revision, we would like to add a clarification in the new Section 2.3 of our reorganized paper where we would compare the two criteria (SOFI and LBWLE):

*When compared to the SOFI index, water-level estimation using landmarks and LBWLE is at a disadvantage because of the necessary and time-consuming ground-survey of the location observed by the camera. Furthermore, landmarks can mostly only be used when the river is out-of-bank, so the approach is not likely to capture drought events. However, the main advantage of this approach compared to SOFI is that it allows estimation of quantitative river levels in accepted units of length (e.g., m). The SOFI index values are dimensionless percentages and to convert them to a height measurement an appropriate scaling must be obtained by calibration with independent data.*

Please also see our answer to Reviewer 1's first comment regarding the alternative ways to obtain topographic data (https://doi.org/10.5194/hess-2021-20-AC1).

**Comment #1c Changes to the title of our paper**

Following the reviewer's comment, we would suggest the following title change to our paper:

*Deep learning for automated river-level monitoring through river camera images: an approach based on water segmentation and transfer learning*

**Comment #1d Emphasis on water-level segmentation in Section 1- Introduction**

If we are invited to revise the manuscript, we will add an additional material about transfer learning to the introduction (see response to Comment #1a). In our view it is necessary to also discuss other types of water level observation and hydrological uses of river cameras in the introduction (see response to Reviewer 1 Comment 4).

**Comment #2a: Suggestions on Language and Writing style - use of first person "we"**

If we are invited to revise the manuscript, it will be checked and some sentences will be rephrased to address this comment.

**Comment #2b: Separate introduction and transfer learning background sections**

If we are invited to revise the paper we will introduce some material on transfer learning in Section 1. In order to fit the typical HESS organization, we will merge Section 2 and 3 into a single Methodology section (see response to Comment #1a and the new table of contents

presented at the end). Some of the ideas on transfer learning from former Section 2 will be rephrased and retained as part of a "Definitions" subsection of our new merged Methodology section. This is to avoid Section 1 becoming excessively long while explaining the important concepts used in our work so that they can be readily understood by HESS readers.

**Comment #3: Check your references**

Vörösmarty et al (2001) include a substantial section on the decline of river gauge data worldwide. We intend to add two more recent references that provide further evidence on this point (Mishra and Coulibaly, 2009; Global Runoff Data Center, 2016).

*Mishra, A.K. and Coulibaly, P., 2009.* Developments in hydrometric network design: A review. *Reviews of Geophysics, 47*(2).

*Global Runoff Data Center, 2016,* Global Runoff Data Base, temporal distribution of available discharge data, [https://www.bafg.de/SharedDocs/Bilder/Bilder_GRDC/grdcStations_tornadoChart.jpg](https://www.bafg.de/SharedDocs/Bilder/Bilder_GRDC/grdcStations_tornadoChart.jpg), *Last accessed: 15 March 2021*

We would rephrase the presentation of the Moy de Vitry et al (2019) paper (line 119):

*Moy de Vitry et al (2019) used a deep semantic segmentation network trained from scratch on an image dataset annotated with water labels. They post-processed the network output in order to produce a generic algorithm for flood level trend monitoring. The SOFI index that they introduced in their post-processing step is related to the percentage of pixels in the image that are estimated as water pixels by the network . This non-dimensional index allows the authors to monitor the evolution of water-levels in their datasets. In our experiments, we will use the SOFI index to track water-level changes (see Section 4).*

We would also give the mathematical expression for the SOFI index in the new Section 2.3 of the reorganized manuscript following Moy de Vitry et al. (2019).

**Comment #4: Give an explanation when computer science terms first appear, like "fine-tune"**

We would like to clarify the explanation of fine-tuning that was given after its first appearance at line 142. Note that given this comment (as well as the restructuring of the manuscript proposed in our responses to comments #1a, #5 and #6), we propose to:

(1) Insert the explanation of fine-tuning in the new Section 2.2.3:

*In Vandaele et al. (2020), the most successful approach considered for applying transfer learning to the semantic segmentation networks is fine-tuning: with fine-tuning, the filter weights obtained by training the network over the source problem are used as initial weights for training the network over the target problem.*

(2) Avoid mentioning fine-tuning in the former Section 3.3 (reorganized section 2.2.2) and use a more generic term instead: the application of transfer learning to train the networks.

(3) Rename Section 3.4 (reorganized section 2.2.3): *Applying transfer learning to train the networks.*

**Comment #5: Explanation of the two transfer learning approaches**

We explained the two transfer learning approaches in former Section 3.4. We hope that these explanations will be more prominent and easier to follow in our reorganized manuscript, where they will appear in section 2.2.3.

**Comment #6: Network structure change for binary segmentation**

We talk about the change in dimension due to the switch to binary segmentation in our transfer learning approach in the former Section 3.4 of our paper. Similarly to Comment #5 and #4 , we hope that the reorganized version of the manuscript could help to avoid the confusion.

**Comment #7: Table 3 metrics**

These metrics are widely used for evaluating flood extent models (e.g. Stephens et al, 2014). However, we understand that including them all in the main paper may not be useful for all readers. We propose to remove the $F^1$, $F^2$, $F^3$ and $F^4$ scores from the main paper, but show these in Appendices or Supplementary Materials.

**Comment #8: LBWLE captures floods rather than droughts**

See response to Comment #1b.

**Comment #9: Purpose of setting-up two experiments**

If our manuscript is accepted for revision, we propose to make the following modifications in the introduction of Section 4:

*Two experiments were carried out with this study.*

*The first experiment presented in Section 3.1 is designed to address the suitability of our approach for the automatic derivation of water level observations using river cameras images and landmarks from a ground survey. Landmarks and associated manually derived water-levels are available for a two-week flood event (Vetra-Carvalho et al, 2020). These data allow us to validate our LBWLE approach for water-level estimation in accepted units of length (m) with co-located water-levels estimated by a human observer.*

*With the second experiment presented in Section 3.2, our approach is applied to larger, one year, datasets of camera images that include a larger range of river flow rates and stages. This experiment allows us to better understand the suitability and robustness of the LBWLE and SOFI water-level measurements. However, manually derived co-located water levels are not available for this period, so we use the nearest available river gauge data for validation. For some of the cameras, the nearest gauge is several km away.*

**New Table of Contents:**

---

## Author Response (AR1)

**HESS-2021-20**
*Vandaele et al.* **Deep learning for automated river-level monitoring through river camera images: an approach based on water segmentation and transfer learning**

Thank you for considering our paper for revision. You will find below a point-by-point answer to the reviews. They correspond to slightly adapted versions of our answers made during the interactive process (https://doi.org/10.5194/hess-2021-20-AC1 and https://doi.org/10.5194/hess-2021-20-AC2). The small differences between the answers are related to minor changes that we deemed necessary when we applied our proposed modifications.

The legend of the marked-up version of our revised manuscript is as follows:
- Red: additions to the manuscript
- Blue: modifications to the manuscript
- Green: text & sections that were moved (if parts were moved and modified, blue has the priority)

Note that, unless explicitly stated otherwise, section numbers in our response reflect the new organization of the manuscript that is presented at the end of this document.

**Response to Reviewer #1**

**Comment #1: Regarding the use of alternative ways to obtain topographic data**

We agree that obtaining ground surveys for the field-of-view for each camera is a significant challenge. An alternative could be to use the camera images in conjunction with a high resolution digital surface model (DSM). An open access lidar DSM is available across the UK. Over other parts of the world, it might be necessary to use a DSM that is commercially available (for example the 12m WorldDEM), but the accuracy of the results using a low resolution DSM would need to be carefully evaluated.

We added the following sentence in the conclusion section:

"*Future work will focus on the merging of the water segmentation results with lidar digital surface model (DSM) data available at 1m resolution over the UK (Environment Agency, 2017). This would allow the water segmentation algorithms to provide a direct estimate of the water levels in the areas that are studied, without requiring any ground-surveys.*"

**Comment #2: Regarding camera movement**

From our inspection of the datasets, we found that the camera movement was negligible (a maximum of 2-3 pixels, even on objects far away from the camera). We were especially careful to choose landmark pixel locations that were not too close to the edge of the landmarked object in order to avoid confusions. We would expect typical intensity-based image registration algorithms to deteriorate the results due to changes

in image illumination and movement within the image (of boats, wildlife and debris), although we have not tested this.

We added the following sentence at the end of Section 3.1.1 (Dataset presentation):

"*An inspection of the datasets and results showed that the impact of camera movement was negligible. Machine-Learning based landmark detection algorithms (e.g, Vandaele et al., 2018) could have been used otherwise, but they are unnecessary in the context of this study.*"

**Comment #3: Regarding the use of area features instead of landmarks**

The starting point of this study was to show how well we were able to automate the annotation process in comparison with the manual landmark annotation approach used in our former study (Vetra-Carvalho et al. 2020). We agree that the use of landmarked areas as opposed to landmarked pixels could strengthen the robustness of our results, but it would have required a new and more complex ground survey that was not in the scope of this study.

To address this comment, we have added the following sentence at the end of Section 3.1.1, after the modification of Comment #2:

"*Also note that this work focuses on a simple process relying on single pixel landmark locations annotated by Vetra-Carvalho et al., (2020b). The use of landmarked areas of multiple pixels sharing the same height could likely help to increase the detection performance and should be considered for an optimal use of this landmark-based approach.*"

**Comment #4: Literature**

We added citations to the literature mentioned by the reviewer at line 41 of the introduction section as follows:

"*There have been a number of citizen science projects that investigated the use of crowdsourced observations of river level (e.g. Royem et al., 2012; Lanfranchi et al., 2014; Etter et al., 2020; Lowry et al., 2019; Walker et al., 2019; Baruch, 2018).*"

In our introduction section, we added the following paragraph regarding the use of river cameras:

"*Several studies have already attempted to use videos and still camera images in order to observe flood events. Surface velocity fields can be computed using videos (e.g., Muste et al., 2008; Le Boursicaud et al.,2016; Creutin et al. 2003; Perks et al., 2020). Still images can be used to observe the water-levels, either manually (e.g., Royem et al, 2012; Schoener, 2018; Etter et al, 2020) or automatically, for example by considering image processing edge detection techniques (Eltner et al. 2018). Under the right conditions, these automated water-level estimation techniques can provide good*

*accuracy with uncertainties of only a few mms  (Gilmore et al., 2013; Eltner et al. 2018). However, the performance of these approaches lacks  portability (Eltner et al.,2018.). "*

**Response to Reviewer #2**

For clarity, we have introduced some extra numbering (1a, 1b, 1c etc.) to address the separate points made in each comment.

As part of our response to the comments,  we revised the organization of our manuscript. The new *table of contents* is located at the end of this document. In addition to the changes proposed below, we intend to update parts of the text to better fit this new organization (section introductions for example). For the sake of clarity, we preferred to keep these minor changes out of this answer but they are given in colour in our marked-up version of the revised manuscript (see above for legend).

**Comment #1a: Emphasize the scientific significance of your research, including the transfer learning aspects**

In order to highlight the novelty of our manuscript and answer the reviewer's comment, we made the following changes:

(1) Modify the end of the introduction (starting line 58, ending line 69) to give a better introduction to transfer learning and our previous paper:

*"Over the last decade, transfer learning (TL) techniques have become a common tool to try to overcome the lack of available data (Reyes et al, 2015; Sabbatelli et al, 2018). The aim of these techniques is to repurpose efficient machine learning models trained on large annotated datasets of images to new related tasks where the availability of annotated datasets is much more limited (see Section 2 for more details). Vandaele et al (2020) successfully analysed a set of TL approaches for improving the performance of deep water segmentation networks. This paper builds on the work of Vandaele et al (2020) and studies the performance of these water segmentation networks for the automation of river-level estimation from river-camera images, in the context of flood-related studies. In particular, this work carries out novel experiments realised with new river-camera datasets and metadata that consider the use of several methods to extract quantitative water-level observations from the segmented river-camera images."*

(2) Create a new Section 2 that would be our Methodology section and would encompass:
   ● the former Section 2 introducing semantic segmentation, deep learning and transfer learning
   ● the former Section 3 presenting the application of  transfer learning
   ● the presentation of SOFI and the new LBWLE method moved from former sections 4.1.4 and 4.2.2

We think that this new organization will help the reader to understand our research more clearly.

**Comment #1b Importance and novelty of LBWLE**

Our goal with LBWLE was to propose a way to provide quantitative water-level observations in accepted units (m), using the landmark-annotated dataset at our disposal.

We added a clarification in Section 2.3.3 to compare the two criteria (SOFI and LBWLE):

*When compared to the SOFI index, water-level estimation using landmarks and LBWLE is at a disadvantage because of the necessary and time-consuming ground-survey of the location observed by the camera. Furthermore, landmarks can mostly only be used when the river is out-of-bank, so the approach is not likely to capture drought events. However, the main advantage of this approach compared to SOFI is that it allows estimation of quantitative river levels in accepted units of length (e.g., m). The SOFI index values are dimensionless percentages and to convert them to a height measurement an appropriate scaling must be obtained by calibration with independent data.*

See our answer to Reviewer 1 Comment #1 regarding the alternative ways to obtain topographic data.

**Comment #1c Changes to the title of our paper**

Following the reviewer's comment, we made the following title change to our paper:

*Deep learning for automated river-level monitoring through river camera images: an approach based on water segmentation and transfer learning*

**Comment #1d Emphasis on water-level segmentation in Section 1- Introduction**

We added an additional material about transfer learning to the introduction (see response to Comment #1a). In our view it is necessary to also discuss other types of water level observation and hydrological uses of river cameras in the introduction (see response to Reviewer 1 Comment #4).

**Comment #2a: Suggestions on Language and Writing style - use of first person "we"**

The manuscript was checked and some sentences were rephrased to address this comment.

**Comment #2b: Separate introduction and transfer learning background sections**

We introduced some material on transfer learning in Section 1. In order to fit the typical HESS organization, we merged former Section 2 and 3 into a single Methodology section (see response to Comment #1a and the new table of contents presented at the end). Some of the ideas on transfer learning from former Section 2 were rephrased and retained as part of a "Definitions" subsection of our new merged Methodology section. This is to avoid Section 1 becoming excessively long while explaining the important concepts used in our work so that they can be readily understood by HESS readers.

**Comment #3: Check your references**

Vörösmarty et al (2001) include a substantial section on the decline of river gauge data worldwide. We added two more recent references that provide further evidence on this point (Mishra and Coulibaly, 2009; Global Runoff Data Center, 2016).

*Mishra, A.K. and Coulibaly, P., 2009.* Developments in hydrometric network design: A review. *Reviews of Geophysics*, 47(2).

*Global Runoff Data Center, 2016,* Global Runoff Database, temporal distribution of available discharge data, https://www.bafg.de/SharedDocs/Bilder/Bilder_GRDC/grdcStations_tornadoChart.jpg, *Last accessed: 15 March 2021*

We rephrased the presentation of the Moy de Vitry et al (2019) paper (line 119) and moved it to new Section 2.3.2:

*The experiments presented in this work use the SOFI index to track water-level changes. Moy de Vitry et al. (2019) introduced the SOFI index to extract flood level information from a deep semantic segmentation network trained from scratch on an image dataset annotated with water labels. The SOFI index is related to the percentage of pixels in the image that are estimated as water pixels by the network, as*

$$SOFI = \frac{\#Pixels_{Flooded}}{\#Pixels_{Total}} \quad (1)$$

*This non-dimensional index allows the authors to monitor the evolution of water-levels in their datasets, and can be computed on the entire water mask or only a sub-region.*

Eq 1 corresponds to the mathematical expression for the SOFI index following its definition in Moy de Vitry et al. (2019).

**Comment #4: Give an explanation when computer science terms first appear, like "fine-tune"**

We clarified the explanation of fine-tuning that was given after its first appearance at line 142. Note that given this comment (as well as the restructuring of the manuscript proposed in our responses to comments #1a, #5 and #6), we made the following changes:

(1) Insert the explanation of fine-tuning in the new Section 2.2.3:

*In Vandaele et al. (2020), the most successful approach considered for applying transfer learning to the semantic segmentation networks is fine-tuning: with fine-tuning, the filter weights obtained by training the network over the source problem are used as initial weights for training the network over the target problem.*

(2) Avoid mentioning fine-tuning in the former Section 3.3 (reorganized section 2.2.2) and use a more generic term instead: the application of transfer learning to train the networks.

(3) Rename Section 3.4 (reorganized section 2.2.3): *Applying transfer learning to train the networks.*

**Comment #5: Explanation of the two transfer learning approaches**

We explained the two transfer learning approaches in former Section 3.4. We hope that these explanations will be more prominent and easier to follow in our reorganized manuscript, where they appear in section 2.2.3.

**Comment #6: Network structure change for binary segmentation**

We talk about the change in dimension due to the switch to binary segmentation in our transfer learning approach in the Section 2.2.3 of our paper. Similarly to Comment #5 and #4, we hope that the reorganized version of the manuscript could help to avoid the confusion.

**Comment #7: Table 3 metrics**

These metrics are widely used for evaluating flood extent models (e.g. Stephens et al, 2014). However, we understand that including them all in the main paper may not be useful for all readers. We removed the $F^1$, $F^2$, $F^3$ and $F^4$ scores (and their mentions) from the main paper.

**Comment #8: LBWLE captures floods rather than droughts**

See response to Comment #1b.

**Comment #9: Purpose of setting-up two experiments**

We made the following modifications in the introduction of our new Section 3 (Experiments):

*Two experiments were carried out with this study.*

*The first experiment presented in Section 3.1 is designed to address the suitability of our approach for the automatic derivation of water level observations using river cameras images and landmarks from a ground survey. Landmarks and associated manually derived water-levels are available for a two-week flood event (Vetra-Carvalho et al, 2020). These data allow us to validate our LBWLE approach for water-level estimation in accepted units of length (m) with co-located water-levels estimated by a human observer.*

*With the second experiment presented in Section 3.2, our approach is applied to larger, one year, datasets of camera images that include a larger range of river flow rates and stages. This experiment allows us to better understand the suitability and robustness of the LBWLE and SOFI water-level measurements. However, manually derived co-located water levels are not available for this period, so the nearest available river gauge data for validation was used instead. For some of the cameras, the nearest gauge is several km away.*

**New Table of Contents:**

1. **Introduction** *[former Section 1]*
2. **Transfer learning for water segmentation and river-level estimation**
   *New section encompassing former sections 2 and 3, and addition of a new section concerning SOFI and LBWLE.*
   2.1 Definitions *[former Section 2]*
         2.1.1 Water segmentation for water level estimation [*former Section 2.1*]
         2.1.2 Deep Learning for automated water segmentation *[former Section 2.2]*
         2.1.3 Transfer Learning *[former Section 2.3]*
   2.2 Transfer Learning for deep water semantic segmentation networks *[former section 3]*
         *Former Section 3.1 is removed to avoid repetition of material*
         2.2.1 Network architectures and source datasets *[former Section 3.2]*
         2.2.2 Target datasets for water semantic segmentation *[former Section 3.3]*
         2.2.3 Applying transfer learning to train the networks [*former Section 3.4]*
         2.2.4 Networks retained for the experiments [part of former Section 3.4]
   2.3 River-level estimation using water segmentation [*New section]*
         2.3.1 Static observer flooding index (SOFI) [*former part of Section 4.2.2*]
         2.3.2 Landmark-based water-level estimation (LBWLE) [former part of Section 4.1.4]
         2.3.3 Comparison of SOFI and LBWLE [*new*]
3. **Experiments** *[former Section 4]*
   3.1 Application on a practical case for flood observation *[former Section 4.1]*
         3.1.1 River camera datasets for a flood event on the river Severn and the river Avon *[former Section 3.1.1]*
         3.1.2 Evaluation Protocol *[former Section 4.1.2]*
         3.1.3 Landmark classification results *[former Section 4.1.3]*
         3.1.4 Estimating the water-level using the landmark classification [*former Section 4.1.4]*
   3.2 Performance evaluation for year long water-level analysis *[former Section 4.2]*
         3.2.1 Year-long river-camera images datasets *[former Section 4.2.1]*
         3.2.2 Evaluation protocol *[former Section 4.2.2]*
         3.2.3 Landmark-based water-level estimation analysis *[former Section 4.2.3]*
         3.2.4 Full image SOFI index analysis *[former Section 4.2.4]*
         3.2.5 Windowed image SOFI index analysis *[former Section 4.2.5]*
4. **Conclusion** *[former Section 5]*

---

## Author Response (AR2)

**HESS-2021-20**
*Vandaele et al.* **Deep learning for automated river-level monitoring through river camera images: an approach based on water segmentation and transfer learning**

We thank the reviewer for their useful comments.

Please find our detailed answer here below. The changes to the manuscript are presented in blue font in the answer and the marked up version of the manuscript..

**(a) Add some experiments**
**The necessity of introducing transfer learning into the study has not been fully explored in the paper, which would undermine the scientific significance of your research. It is necessary to set up a control group without using transfer learning method to illustrate the significance of transfer learning. And the comparison between this experiment and the two experiments using transfer learning should be illustrated in table or figure. Through the comparison, the scientific significance of the transfer learning in this task will be further clarified.**

We would like to note that the transfer learning aspect of our semantic segmentation approach (usefulness, comparison with state-of-the-art) was studied in a previous work published at a computer vision conference (Vandaele et al., 2020). We have made the following modifications presented in the introduction:

*Vandaele et al., 2020 successfully analysed a set of TL approaches for improving the performance of deep water segmentation networks by showing that they could outperform water segmentation networks trained from scratch over the same datasets. This paper builds on the work of Vandaele et al., 2020 and studies the performance of these water segmentation networks trained using TL approaches for the automation of river-level estimation from river-camera images, in the context of flood-related studies. In particular, this work uses water segmentation networks trained using TL approaches in order to carry out novel experiments realised with new river-camera datasets and metadata that consider the use of several methods to extract quantitative water-level observations from the water-segmented river-camera images.*

**(b) The focus of the introduction**
**Keeping the description on other types of water level observation and hydrological uses of river cameras in the introduction part is surely no problem, it is just the matter of length. Since the novelty of the work is the modification on the existing deep learning methods rather than to propose a new model to replace the traditional observation methods, it is still recommended that you emphasize more on the existing computer vision methods for water segmentation. From line 58 to line 65, you can add more reference including research on histogram analysis and machine learning methods for water segmentation. The difference and commonality of these methods are also worth introducing. These computer vision methods are the basis of your work, they deserve more description.**

We have made following changes to the paragraph from line 58 to 65:

*By extracting the location of water-filled pixels from a stream of river camera images (water segmentation), it becomes possible to analyse flood events happening within the field-of-view of a camera. Most attempts that have tried to tackle the problem of automated water detection in the context of floods have been realised through the histogram analysis of the image (Filonenko et al., 2015; Zhou et al., 2020). These algorithms remain sensitive to luminosity and water reflection problems (Filonenko et al., 2015) unless the dynamic aspect of the video feed can be exploited (e.g., 25fps in Mettes et al., 2014) or the camera is set to observe a specific gauge/ruler (Pan et al., 2018), which is not the case for the river cameras used in this work (1 frame per hour). Deep learning approaches have been applied to flood detection using river cameras (Lopez-Fuentes et al., 2017; Moy de Vitry et al., 2019). However, current flood-related studies using river camera images are limited because the observations made on the stream of images must be annotated manually (Vetra-Carvalho et al., 2020b). An accurate, manual annotation of such images is a long and tedious process that compels the analyst to narrow the scope (number of images considered) of the study.*

**Newly added references:**

Zhou, S., Kan, P., Silbernagel, J., & Jin, J. (2020). Application of image segmentation in surface water extraction of freshwater lakes using radar data. *ISPRS International Journal of Geo-Information*, 9(7), 424. **https://doi.org/10.3390/ijgi9070424**

Mettes, P., Tan, R. T., & Veltkamp, R. (2014, January). On the segmentation and classification of water in videos. In *2014 International Conference on Computer Vision Theory and Applications (VISAPP)* (Vol. 1, pp. 283-292). IEEE. **https://doi.org/10.13140/2.1.2141.2809**

Pan, J., Yin, Y., Xiong, J., Luo, W., Gui, G., & Sari, H. (2018). Deep learning-based unmanned surveillance systems for observing water levels. *IEEE Access*, 6, 73561-73571. **https://doi.org/10.1109/ACCESS.2018.2883702**

**(c) The details about the CNN model**
**A CNN based model is typically composed of three types of layers, which are convolutional layer, pooling layer and fully-connected layer. However, in Section 2.1.2, only the computational process in convolutional layer is introduced. Are the other two layers used in the model in your work? If so, please supplement the introduction of them, and revise Figure 2 according to the added introduction. This will also help readers to understand the main difference between the two transfer learning strategies in Section 2.2.3 and the scientific significance of the work on comparing different strategies. Additionally, it is also necessary to simply mention the activation function between different layers in CNN model, e.g., ReLu or Softmax.**

As explained in L85, our goal with the brief explanation of the CNN was to present the concepts of the CNN to a community that is not oriented towards deep learning so that anyone could read the paper without referring to other work. The aim is not to describe the architectures that we used (Resnet50-Upernet and Deeplab). However, we agree that some details could help  the reader's understanding, so we have made the following changes:

Paragraph starting line 97:
*As for most image-processing related tasks, recent advances in optimisation, parallel computing and dataset availability have allowed deep learning methods, and specifically deep convolutional neural networks (CNNs) to bring major improvements to the field of*

*automated semantic segmentation (Guo et al., 2018). CNNs are a type of neural network where input images are processed through convolution layers. As it is shown in Fig. 2, with convolutional neural networks, an image is divided into square sub-regions (tiles) of size F × F that can possibly overlap. The image is processed through a series of convolutional layers. A convolutional layer is composed of filters (matrices) of size $F \times F \times C_i$, where C is the number of channels of the input image at layer i . For each filter of the convolutional layer, the filter is applied on each of the tiles of the image by computing the sum of the Hadamard product (element-wise matrix multiplication) - also called a convolution in deep learning - between the tile and the filter (Strang, 2019), which is then processed through an activation function (e.g. ReLU (Nair and Hinton, 2010), sigmoid or identity function). If the products of the convolution operations are organised spatially, the output of a convolutional layer can be seen as another image which itself can be processed by another convolutional layer: if a convolutional layer is composed of N filters, then the output "image" of this convolutional layers has N channels. CNN architectures vary in number of layers and choice of activation function, but also in terms of additional layers. Typically, SoftMax layers are added at the end of categorization/classification tasks (such as semantic water segmentation) to normalize the last $C_l$ channels into a probability distribution of $C_i$ category/classes. Pooling layers are often used to reduce the dimension of a layer by computing the maximum (max-pooling)/average (average-pooling) of partitions (non-overlapping contiguous regions) of size P x P of the input image.*

*During the training of the networks, the weights of the filters (the matrix values) are optimised. The idea is that the filters will converge along the convolutional layers towards weights making the input image more and more meaningful for the task at hand.*

**Newly added references:**
Vinod Nair et Geoffrey E. Hinton, « Rectified linear units improve restricted boltzmann machines », *Proceedings of the 27th International Conference on Machine Learning (ICML-10)*, 2010

Paragraph starting line 163:
*The semantic segmentation networks that were chosen are addressing semantic segmentation problems with 171 (COCO-stuff) and 150 (ADE20k) labels (see Section 2.2.1) and use a SoftMax layer (see Section 2.1.2) to perform their segmentation, which means that their last layer has as many filter as there are labels. However, the water semantic segmentation problem is a binary segmentation problem, with only two labels: water or not-water. In practice, this means that the last layer of the source semantic segmentation networks and the target semantic segmentation networks will have a different number of filters. In consequence, it is not possible to use the weights of the last layer of the source network to initialise the weights of the last layer of the target network. This is why two fine-tuning strategies were considered in Vandaele et al., 2020:*

**(d) The setup of the experiments**
**In Section 3, some key information about model training should be supplemented. Firstly, the setup of hyperparameters is supposed to be added, including learning rate, training epoch, batch size as well as the optimizer for training. Simultaneously, the introduction of learning rate setting would help to understand the meaning of fine-tuning if compared with the learning rate in original training with large dataset. Secondly, the machine learning library used in this study needs to be illustrated,**

**Pytorch, Tensorflow or Keras? Thirdly, which loss function is used for supervising the training should be illustrated, cross entropy loss, MSE loss or others? Fourthly, could you please give some comparison of two training strategies on convergence speed?**

We propose to change the paragraph L179-181 to the following:

*As explained in Vandaele et al. 2020, the training used 300 epochs in order to ensure full convergence for all the networks. The initial learning rate value for the fine-tuning was 10 times smaller than its recommended value (0.001) in order to start with less aggressive updates. The other parameters (loss, update schedule, batch size) were chosen as recommended by the authors of the networks (Zhou et al., 2018; Chen et al., 2017). Both authors implemented their network using the Pytorch library.*